# Pathogenic PS1 phosphorylation at Ser367

**Masato Maesako[1†], Jana Horlacher[1,2†], Katarzyna M Zoltowska[1], Ksenia V Kastanenka[1], Eleanna Kara[1], Sarah Svirsky[1], Laura J Keller[1], Xuejing Li[1], Bradley T Hyman[1], Brian J Bacskai[1], Oksana Berezovska[1]\***

[1]Alzheimer's Disease Research Laboratory, MassGeneral Institute for Neurodegenerative Disease, Massachusetts General Hospital, Harvard Medical School, Charlestown, United States; [2]Department of Neurology, University of Ulm, Ulm, Germany

**Abstract** The high levels of serine (S) and threonine (T) residues within the Presenilin 1 (PS1) N-terminus and in the large hydrophilic loop region suggest that the enzymatic function of PS1/$\gamma$-secretase can be modulated by its 'phosphorylated' and 'dephosphorylated' states. However, the functional outcome of PS1 phosphorylation and its significance for Alzheimer's disease (AD) pathogenesis is poorly understood. Here, comprehensive analysis using FRET-based imaging reveals that activity-driven and Protein Kinase A-mediated PS1 phosphorylation at three domains (domain 1: T74, domain 2: S310 and S313, domain 3: S365, S366, and S367), with S367 being critical, is responsible for the PS1 pathogenic 'closed' conformation, and resulting increase in the A$\beta$42/40 ratio. Moreover, we have established novel imaging assays for monitoring PS1 conformation in vivo, and report that PS1 phosphorylation induces the pathogenic conformational shift in the living mouse brain. These phosphorylation sites represent potential new targets for AD treatment.

**\*For correspondence:**
OBEREZOVSKA@mgh.harvard.edu

[†]These authors contributed equally to this work

**Competing interests:** The authors declare that no competing interests exist.

## Introduction

Senile plaques, comprised primarily of $\beta$-amyloid (A$\beta$), are the major pathological hallmark of Alzheimer's disease (AD). Presenilin 1 (PS1) is the catalytic component of $\gamma$-secretase (*Wolfe et al., 1999*; *Esler et al., 2000*; *Li et al., 2000*), which is responsible for the final enzymatic cleavage of amyloid precursor protein (APP) to generate A$\beta$ (*De Strooper et al., 1998*). Over 180 familial AD mutations have been identified in the PS1 gene, and the majority of them lead to an increase in the A$\beta$42/40 ratio (*De Strooper, 2007*). PS1 knock-in mice carrying familial AD mutations exhibit decreased A$\beta$40 and A$\beta$42 levels but increased A$\beta$42/40 ratio and accelerated A$\beta$ deposition (*Xia et al., 2015*). This supports the idea that up-regulation of the A$\beta$42/40 ratio rather than the total amount of A$\beta$ has a strong impact on A$\beta$ deposition and AD pathogenesis.

Using Förster resonance energy transfer (FRET)-based imaging techniques we have previously shown that familial AD mutations in PS1 increase the proximity of PS1 N-terminus (NT) to C-terminus (CT) or to the large cytoplasmic loop domain, causing the so called 'closed' pathogenic PS1 conformation (*Berezovska et al., 2005*; *Uemura et al., 2009*). On the other hand, several nonsteroidal anti-inflammatory drugs and PS1/$\gamma$-secretase modulators, which are known to decrease the A$\beta$42/40 ratio (*Weggen et al., 2001*; *Page et al., 2008*), drive PS1 into the 'open' conformation (*Lleó et al., 2004*; *Uemura et al., 2009*; *Ohki et al., 2011*). These reports indicate that PS1 conformational changes are tightly linked to changes of the A$\beta$42/40 ratio, suggesting that modulation of the pathogenic 'closed' PS1 conformation is a potential target for AD treatment.

**eLife digest** Alzheimer's disease is a widely recognised disorder caused by the progressive deterioration and death of brain cells. A key feature of the disease is the formation of structures called plaques in the brain. Plaques occur when many copies of a molecule known as amyloid beta stick together outside of the brain cells. Healthy brains also produce amyloid beta but it is in a different form, which cannot form plaques.

One in twenty people with Alzheimer's disease have a family history of the disease. Of these, many are linked to changes in a gene that produces a protein called Presenilin 1 (or PS1 for short). Cells need PS1 to make amyloid beta and the altered versions of PS1 produce the type of amyloid beta that causes Alzheimer's disease. Yet, in cases that do not run in families, the gene for PS1 is unchanged but the PS1 protein still produces the form of amyloid beta that is linked to Alzheimer's disease.

Maesako, Horlacher et al. wanted to find out how seemingly healthy PS1 proteins can be made to produce plaque-forming amyloid betas. Studies of PS1 from mice revealed that small chemical modifications, called phosphate groups, could be attached to PS1 in a process called phosphorylation. Modified PS1 proteins produce harmful amyloid betas and removing the modifications was enough to make PS1 behave normally again. Maesako, Horlacher et al. found three points in the PS1 protein where phosphorylation could change the behaviour of the protein, the most important one is a site called Ser367.

Further investigation showed that an enzyme called Protein Kinase A (PKA) phosphorylates PS1; this enzyme is also able to attach phosphate groups to many different proteins. Maesako, Horlacher et al. went on to show that PS1 is phosphorylated in samples from people with Alzheimer's disease, suggesting that this is a plausible cause for some cases of the disease. Finding a way to prevent phosphorylation or remove phosphate groups from PS1 could be the first step towards treating these cases of Alzheimer's disease.

A number of studies highlight the reciprocal relationship between elevation in intracellular $Ca^{2+}$ levels and $A\beta$ pathology. Notably, $A\beta$ deposition is observed in brain regions exhibiting a high basal rate of neuronal activity in humans (*Sperling et al., 2009*). The idea that $A\beta$ causes $Ca^{2+}$ elevation is supported by several studies (*Kuchibhotla et al., 2008*; *Lopez et al., 2008*; *Busche et al., 2012*; *Higuchi et al., 2012*). On the other hand, aberrant $Ca^{2+}$ homeostasis affects $A\beta$ production. For example, acute stimulation of the entorhinal cortex using an optogenetic approach increases the $A\beta42$ level specifically in the projection area of the perforant pathway (*Yamamoto et al., 2015*). KCl-induced depolarization causes sustained release of $A\beta42$ from AD mouse model synaptoneurosomes (*Kim et al., 2010*). KCl-triggered $Ca^{2+}$ influx elevates intracellular $A\beta42/40$ ratio in primary neurons (*Pierrot et al., 2004*). We recently reported that increases in $Ca^{2+}$ levels induce the PS1 pathogenic 'closed' conformation in primary neurons, followed by an increase in the $A\beta42/40$ ratio (*Kuzuya et al., 2016*). Furthermore, PS1 is found in the 'closed' conformation in sporadic AD brain, and this pathogenic PS1 conformational shift correlates with the proximity to deposited $A\beta$ (*Wahlster et al., 2013*). In this report, we address the detailed molecular mechanisms of the $Ca^{2+}$-driven pathogenic 'closed' conformation of the PS1/γ-secretase.

Phosphorylation, one of the crucial post-translational modifications, is a rapid and dynamic event in cells, and represents a common mechanism of regulating protein conformation and/or activity. In PS1, fifteen phosphorylation sites have been identified (*Kirschenbaum et al., 2001*; *Lau et al., 2002*; *Fluhrer et al., 2004*; *Kuo et al., 2008*; *Ryu et al., 2010*; *Matz et al., 2015*). Although the phosphorylation of some of these sites affects the stability of PS1/γ-secretase (*Lau et al., 2002*; *Kuo et al., 2008*; *Ryu et al., 2010*), the impact of phosphorylation on PS1 conformation and function has not yet been tested. Here, using FRET-based imaging we show that phosphorylation-inhibited mutations at the three domains (domain 1: T74, domain 2: S310, S313, and domain 3: S365, S366 and S367) prevents the $Ca^{2+}$-mediated PS1 'closed' conformational change. Conversely, phosphorylation-mimicking mutations reveal that domain 1 and 2 phosphorylation is necessary, but domain 3, and S367 in particular, is crucial for causing the PS1 pathogenic 'closed' conformation.

Moreover, Protein Kinase A (PKA) inhibitors prevent the $Ca^{2+}$-induced PS1 'closed' conformational change whereas PKA activators trigger it. Therefore, we conclude that the $Ca^{2+}$-increased PKA activity leads to phosphorylation of PS1 at the three domains, inducing the PS1 pathogenic 'closed' conformation. These results provide strong evidence that PS1 phosphorylation at certain residues can potentially be molecular targets for AD prevention.

## Results

### $Ca^{2+}$ influx increases the level of phosphorylated PS1

Using the conformation sensitive FRET probe, GFP-PS1-RFP (G-PS1-R) (*Uemura et al., 2009*), and the real-time FRET assay, we have previously shown that PS1 conformation is dynamically regulated by intracellular $Ca^{2+}$. Specifically, we found that KCl or glutamate-induced increases of intracellular $Ca^{2+}$ levels led to PS1 adopting a pathogenic 'closed' conformation (*Kuzuya et al., 2016*). Now, by employing an antibody based FRET assay, we detect changes in the conformation of endogenous PS1 in primary neurons. We show that treatment with KCl significantly increased the proximity between fluorescently labeled PS1 N-terminus (NT) and PS1 C-terminus (CT) (increased red pixels mean higher FRET efficiency, *Figure 1A*), indicating that KCl treatment triggers the 'closed' conformation of endogenous PS1.

Since phosphorylation is known to regulate protein conformation and/or activity, we tested whether $Ca^{2+}$ influx may induce phosphorylation of PS1, and whether PS1 phosphorylation is responsible for the PS1 pathogenic conformational change. For this, the amount of phosphorylated PS1 was measured by immunoprecipitation/Western blotting in primary neurons treated with KCl. We found that KCl treatment significantly increased the level of PS1 phosphorylation at its serine residues (*Figure 1B*). We have confirmed this finding using another stimulant, glutamate (*Figure 1—figure supplement 1*). Collectively, these results indicate that $Ca^{2+}$ influx leads to PS1 phosphorylation.

### Phosphorylation of 6 sites is necessary for the $Ca^{2+}$-triggered PS1 pathogenic conformational change

To address whether PS1 phosphorylation is involved in the $Ca^{2+}$-mediated PS1 conformational change, we used phosphorylation-inhibited mutants of PS1. For this, 7 W cells were transiently transfected with wild type (WT) or mutant PS1 constructs, in which serine (S) or threonine (T) previously reported as phosphorylation sites were substituted by alanine (A) (*Table 1*). Then, transfected cells were treated with the $Ca^{2+}$ ionophore A23187 to increase intracellular $Ca^{2+}$ levels. Following the treatment, the cells were fixed and immunostained with anti-FLAG (on PS1 NT) and anti-PS1 CT antibodies to selectively detect the conformational change of 'exogenous' PS1 using the antibody-based FRET assay. Significant increases in $Ca^{2+}$ levels were verified by Indo-1 $Ca^{2+}$ imaging (*Grynkiewicz et al., 1985*) (*Figure 2—figure supplement 1A*). There was no statistical difference between the FRET efficiencies of WT PS1 and phosphorylation-inhibited PS1 mutants in vehicle-treated cells (*Table 1*), indicating that the conformation of the phosphorylation-inhibited PS1 mutants is not different from that of WT PS1 in basal conditions. A23187 significantly increased the FRET efficiency between fluorescently labeled NT and CT of WT PS1 (*Table 1*). This indicates that WT PS1 adopts the 'closed' conformation following the treatment with the $Ca^{2+}$ ionophore. On the other hand, A23187 did not change the FRET efficiencies in T74A, S310A, S313A, S310A/S313A, S365A, S366A, S367A, S365A/S367A and S366A/S367A PS1 transfected cells, suggesting that conformation of these PS1 mutants was resistant to $Ca^{2+}$ ionophore treatment (*Table 1*). The A23187-resistant phosphorylation sites on PS1 could be divided into three sub-membrane domains: domain 1 (T74) on NT of PS1, domain 2 (S310, S313) and domain 3 (S365, S366, S367) within the large hydrophilic loop (*Figure 2—figure supplement 2*).

To confirm this finding we used a complementary approach, for which we introduced phosphorylation-inhibited mutations within the three domains into GFP-PS1-RFP, a conformation sensitive FRET probe (*Uemura et al., 2009*). Consistently, A23187 significantly increased the FRET efficiency of WT G-PS1-R (*Figure 2—figure supplement 1B*). On the other hand, A23187 did not change the FRET efficiency between GFP and RFP of domain 1, domain 2, or domain 3 phosphorylation-inhibited G-PS1-R mutants (*Figure 2—figure supplement 1B*). Collectively, these results demonstrate

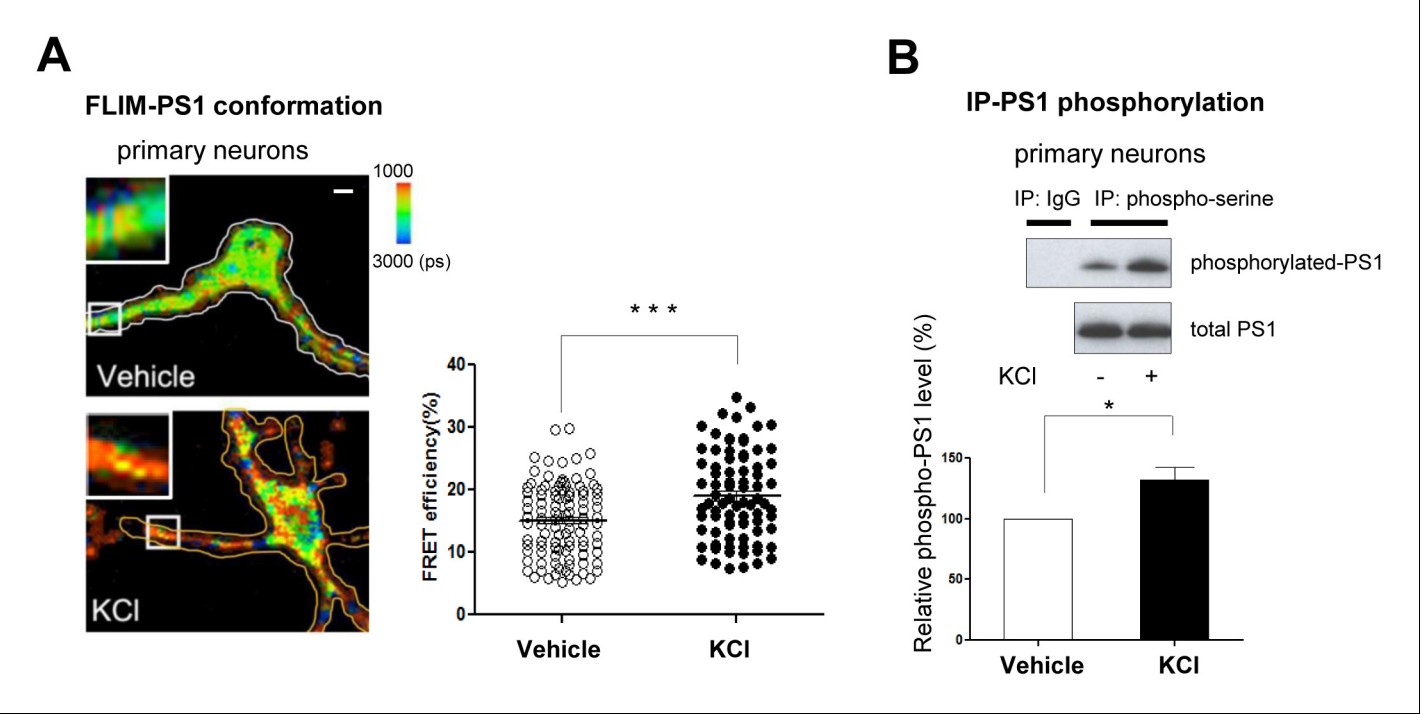

**Figure 1.** Ca$^{2+}$ influx-triggered PS1 conformational change and increased phosphorylation. (**A**) FLIM analysis of PS1 conformation. Pseudo-colored FLIM images of primary neurons treated with vehicle control or 50 mM KCl for 5 min. The neurons were stained with the antibodies to PS1 NT (Alexa488) and PS1 CT (Cy3). The colorimetric scale shows Alexa488 lifetime in picoseconds (ps). A scale bar indicates 10 μm. The graph shows quantitative analysis of the FRET efficiency between fluorescently labeled PS1 NT and PS1 CT in neuronal processes (total of 81–103 processes from 32–38 cells). Mean ± SEM, \*\*\*p<0.001, Student's t-test. (**B**) Immunoprecipitation/Western blot analysis of PS1 phosphorylation. Primary neurons treated with vehicle control or 50 mM KCl for 5 min were immunoprecipitated with mouse and rabbit anti-phosphoserine antibody mixture, followed by immunoblotting with the anti-PS1 loop antibody (Upper panel, top row). Normal mouse and rabbit IgG mixture was used as negative control. Lower gel shows the total level of PS1 CTF in neuronal cell lysates. The graph presents a quantitative analysis of the band intensity for phosphorylated PS1 (n = 6). The relative PS1 phosphorylation level in KCl-treated neurons was normalized to that in vehicle-treated cells. Mean ± SEM, \*p<0.05, One sample t-test.

The following figure supplement is available for figure 1:

**Figure supplement 1.** Glutamate treatment induces PS1 phosphorylation.

that phosphorylation of the three domains is necessary for the Ca$^{2+}$ dependent PS1 pathogenic conformational shift.

## Phosphorylation of PS1 at S367 is crucial for the pathogenic conformational shift

It still remains unknown whether simultaneous phosphorylation of all three domains is necessary for the pathogenic change in PS1 conformation, or if there is a sequence of phosphorylation events leading to the PS1 conformational change. To distinguish between these scenarios, we generated a set of phosphorylation-mimicking G-PS1-R mutants by replacing serine or threonine with aspartic acid (D) in single domains (domain 1, domain 2 or domain 3), in double domains (domains 1 + 2, domains 1 + 3 or domains 2 + 3), or in triple domains 1 + 2 + 3. The conformation of G-PS1-R with domain 1, or domain 2 only, or combined domain 1 and 2 mutants was not different from that of WT G-PS1-R (*Figure 2A*). However, the FRET efficiency of the domain 3 phosphorylation-mimicking G-PS1-R isoform was significantly higher than that of WT G-PS1-R, indicating that this mutant forces PS1 into the 'closed' conformation (*Figure 2A*). Domains 1 + 3 G-PS1-R, domains 2 + 3 G-PS1-R and domains 1 + 2 + 3 G-PS1-R also adopted the 'closed' conformation (*Figure 2A*). Collectively, these results strongly suggest that phosphorylation of domain 3 is the critical mechanism leading to the PS1/γ-secretase pathogenic 'closed' conformation.

**Table 1.** FLIM analysis of the PS1 NT-CT proximity in phosphorylation-inhibited PS1 mutants.

| Construct | *Relative FRET efficiency (%) | | p value | |
| | DMSO | A23187 | ‡vs WT (in DMSO) | §DMSO vs A23187 |
| --- | --- | --- | --- | --- |
| PS1 Wild type (WT) | 100 ± 8.9 (n = 18) | 134.7 ± 7.5 (n = 25) | - | †p<0.05 |
| PS1 S28A | 106.9 ± 7.0 (n = 19) | 143.4 ± 8.6 (n = 19) | n.s. | †p<0.05 |
| PS1 T74A | 96.5 ± 13.9 (n = 14) | 98.5 ± 8.3 (n = 22) | n.s. | n.s. |
| PS1 S310A | 88.8 ± 10.2 (n = 17) | 95.1 ± 8.6 (n = 24) | n.s. | n.s. |
| PS1 S313A | 92.7 ± 8.4 (n = 15) | 108.0 ± 8.8 (n = 21) | n.s. | n.s. |
| PS1 S310A/S313A | 84.2 ± 6.7 (n = 26) | 87.8 ± 9.0 (n = 17) | n.s. | n.s. |
| PS1 S319A/T320A | 106.2 ± 8.1 (n = 20) | 131.8 ± 8.1 (n = 20) | n.s. | †p<0.05 |
| PS1 S324A | 93.8 ± 6.2 (n = 19) | 129.9 ± 8.4 (n = 15) | n.s. | †p<0.05 |
| PS1 S337A | 99.9 ± 8.8 (n = 13) | 132.4 ± 10.9 (n = 16) | n.s. | †p<0.05 |
| PS1 S346A | 100.1 ± 8.9 (n = 18) | 135.0 ± 5.2 (n = 19) | n.s. | †p<0.05 |
| PS1 S353A | 89.6 ± 7.4 (n = 18) | 129.8 ± 8.3 (n = 23) | n.s. | †p<0.05 |
| PS1 T354A | 74.4 ± 5.6 (n = 14) | 113.4 ± 7.7 (n = 23) | n.s. | †p<0.05 |
| PS1 S357A | 91.2 ± 7.9 (n = 18) | 119.8 ± 9.2 (n = 23) | n.s. | †p<0.05 |
| PS1 S353A/S357A | 98.9 ± 9.1 (n = 14) | 125.9 ± 7.1 (n = 13) | n.s. | †p<0.05 |
| PS1 S365A | 102.3 ± 6.4 (n = 19) | 87.3 ± 9.9 (n = 21) | n.s. | n.s. |
| PS1 S366A | 102.8 ± 6.7 (n = 16) | 101.6 ± 12.6 (n = 16) | n.s. | n.s. |
| PS1 S367A | 99.7 ± 6.1 (n = 10) | 99.1 ± 11.4 (n = 15) | n.s. | n.s. |
| PS1 S365A/S367A | 104.1 ± 6.7 (n = 19) | 102.8 ± 8.4 (n = 19) | n.s. | n.s. |
| PS1 S366A/S367A | 102.7 ± 11.2 (n = 10) | 108.2 ± 9.9 (n = 19) | n.s. | n.s. |

*The FRET efficiency in DMSO-treated cells expressing WT PS1 is set as 100%, and relative FRET efficiency in phosphorylation-inhibited mutants of PS1 is shown. Mean ± SEM, Student's t-test, n: cell number, †: p<0.05, n.s.: not significant.

‡p-value is shown for the comparison between WT PS1 and phosphorylation-inhibited mutants of PS1 in DMSO-treated conditions, or §for the comparison between DMSO-treated and A23187 (5 µM for 15 min)-treated cells expressing the same PS1 construct.

Domain 3 of PS1 consists of three consecutive serine residues: S365, S366 and S367. To identify specific residues in domain 3 responsible for the pathogenic conformational change of PS1, we next generated single phosphorylation-mimicking G-PS1-R mutants. The spectral FRET analysis revealed significantly higher FRET efficiency in S367D G-PS1-R but not in S365D G-PS1-R or S366D G-PS1-R, as compared to WT G-PS1-R (**Figure 2B**). This indicates that phosphorylation of PS1 at serine 367 is responsible for the PS1 'closed' conformational change.

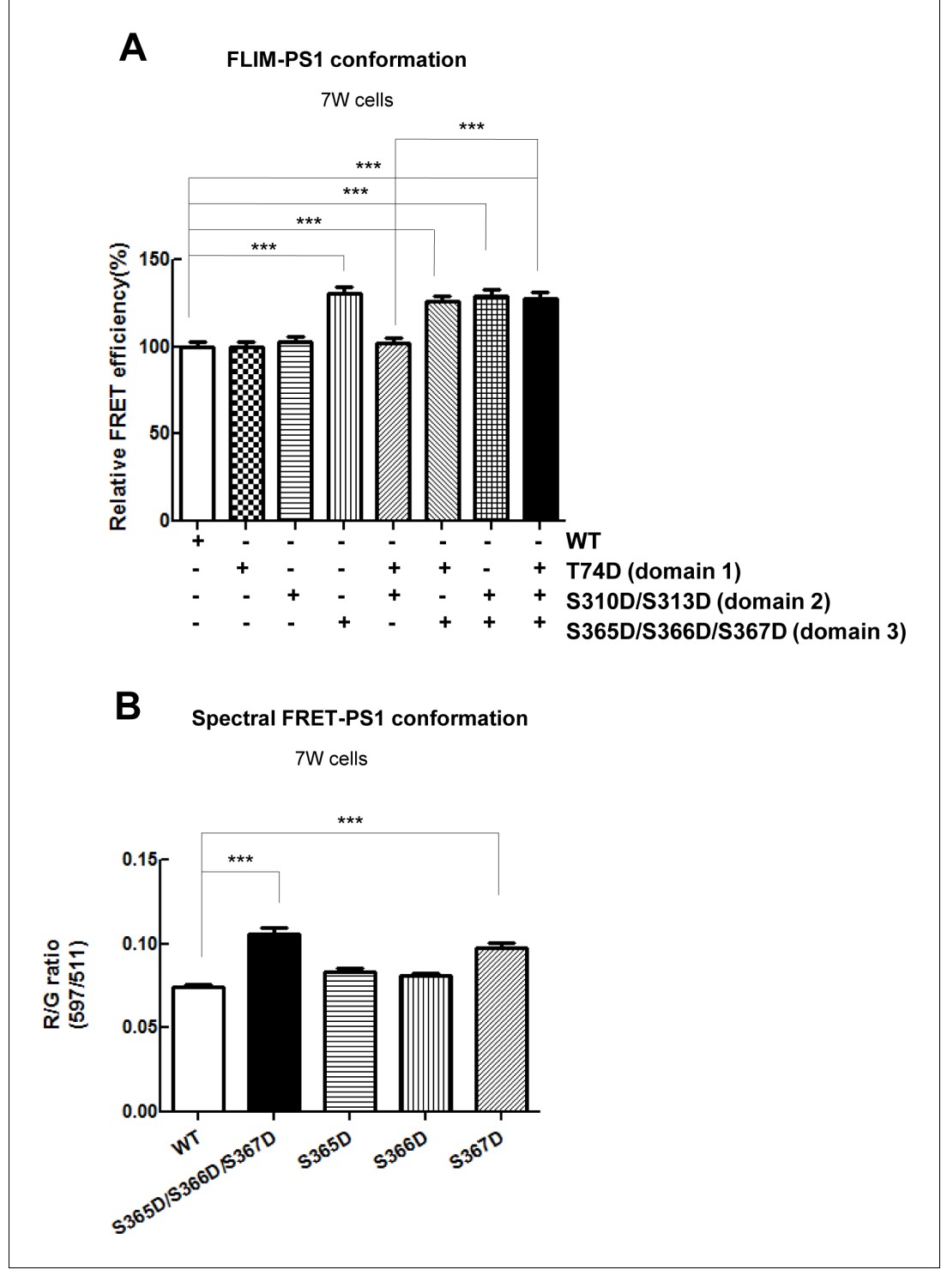

**Figure 2.** Phosphorylation mimicking mutations at S365-S366-S367 result in the PS1 conformational shift. (**A**) FLIM analysis of the PS1 conformation in 7 W cells transfected with WT or phosphorylation-mimicking mutant G-PS1-R. The FRET efficiency between GFP and RFP in phospho-mutants is normalized to the average FRET efficiency of the WT G-PS1-R expressing cells (n = 42–53 cells). Mean ± SEM, ***p<0.001, one-way factorial ANOVA. (**B**) Spectral FRET analysis shows RFP/GFP (R/G) ratio in 7 W cells transfected with WT or single phosphorylation-mimicking mutant G-PS1-R (n = 50–69 cells). Mean ± SEM, ***p<0.001, one-way factorial ANOVA.

The following figure supplements are available for figure 2:

*Figure 2 continued on next page*

*Figure 2 continued*

**Figure supplement 1.** Calcium imaging and validation of the phosphorylation at the three domains-driven PS1 conformational change by the G-PS1-R FRET reporter probe.

**Figure supplement 2.** Schematic image of the three domains in PS1 involved in $Ca^{2+}$-triggered pathogenic 'closed' conformation.

## PKA activity is involved in the $Ca^{2+}$-triggered PS1 pathogenic conformational change

Several kinases involved in phosphorylation of the serine residues within the PS1 cytoplasmic loop region, such as PKA, JNK, PKC and GSK3$\beta$, have been identified (*Kirschenbaum et al., 2001*; *Fluhrer et al., 2004*; *Kuo et al., 2008*). However, the kinase(s) responsible for the phosphorylation that is crucial for the PS1 conformational change remains unknown. To address this, WT G-PS1-R expressing 7 W cells were incubated with inhibitors of several kinases prior to A23187 treatment. The spectral FRET assay revealed that only the PKA inhibitor (KT5720) specifically rescued the $Ca^{2+}$-triggered 'closed' PS1 conformation, whereas JNK, PKC and GSK3$\beta$ inhibitors did not (*Figure 3A*). This was confirmed using a different PKA inhibitor (H-89) and a complementary antibody-based FLIM assay (*Figure 3—figure supplement 1A*). To verify this further in primary neurons, we monitored the change of endogenous PS1 conformation following pre-treatment with the two PKA inhibitors: H-89 or KT5720, or PKA activators: forskolin or 8-Bromo-cAMP, prior to incubation with KCl. The cAMP response element binding protein (CREB), a well known PKA-specific substrate (*Gonzalez and Montminy, 1989*), was used as a positive control for the $Ca^{2+}$ influx-induced PKA activation (*Ferguson and Storm, 2004*) (*Figure 3—figure supplement 1B*). We found that the proximity between the fluorescently labelled PS1 NT and CT was comparable between vehicle and KCl-treated neurons in the presence of H-89 or KT5720 (*Figure 3B*). On the other hand, the FRET efficiency of forskolin or 8-Bromo-cAMP treated neurons was significantly higher than that of vehicle-treated neurons (*Figure 3B*), supporting the involvement of PKA in the PS1 conformational change. Of note, there was no additive effect with KCl and PKA activators (*Figure 3B*). The inhibitory effect of KT5720 on the KCl/$Ca^{2+}$-triggered PS1 conformational change was also confirmed by real-time spectral FRET imaging in living primary neurons (*Figure 3—figure supplement 1C*). To further corroborate the results obtained by pharmacological inhibition of PKA, we employed a dominant negative form of the PKA regulatory α subunit (G324D) (*Olson et al., 1993*). We found that co-expression of the G324D PKA mutant together with G-PS1-R prevents both $Ca^{2+}$-triggered PS1 phosphorylation at S310 and the 'closed' PS1 conformation (*Figure 3—figure supplement 1D*).

To further verify that the $Ca^{2+}$-induced PKA activation is an essential step in the PS1 conformational change, we pre-treated primary neurons with forskolin or 8-Bromo-cAMP, followed by treatment with EGTA to reduce intracellular $Ca^{2+}$. As expected, EGTA decreased the FRET efficiency between PS1 NT and CT, indicating that PS1 adopted an 'open' conformation (*Figure 3C*). On the other hand, EGTA had no effect on PS1 conformation in the presence of PKA activators (*Figure 3C*), indicating that PKA activation is an event down-stream of the $Ca^{2+}$ influx.

To ensure that PKA activation causes the PS1 pathogenic conformational change via PS1 phosphorylation, we treated 7 W cells expressing WT G-PS1-R or the phosphorylation-inhibited G-PS1-R mutants with forskolin. Whereas PKA activation by forskolin caused the PS1 'closed' conformation in WT G-PS1-R expressing cells, it did not affect the conformation of domain 1, domain 2, or domain 3 phosphorylation-inhibited mutants (*Figure 3D*). This indicates that blocking phosphorylation of either one of these domains precludes PS1 from adopting the 'closed' conformation via PKA activation.

Phosphorylation-mimicking mutations in domain 1 or domain 2 only (but not in domain 3), or domain 1 and domain 2 combined had no effect on PS1 conformation (*Figure 2A*). Thus, to verify that domain 3 needs to be phosphorylated for the PS1 conformational change, and to determine whether PKA is involved, we transfected 7 W cells with WT or the domains 1 + 2 phosphorylation-mimicking G-PS1-R mutant, followed by treatment with A23187, forskolin, or 8-Bromo-cAMP. Similar to WT G-PS1-R expressing cells, the FRET efficiency of domains 1 + 2 phosphorylation-mimicking

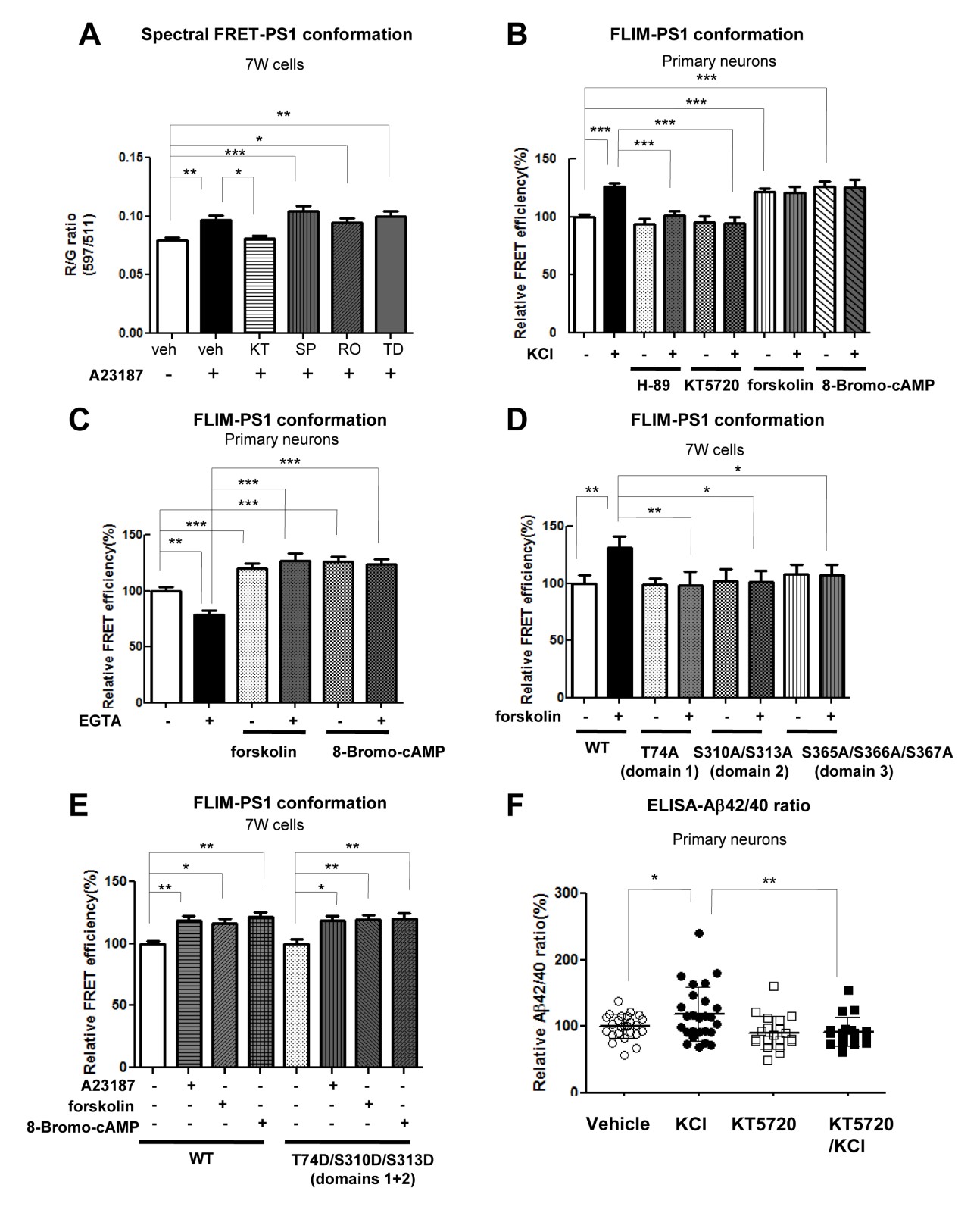

**Figure 3.** PKA activity is involved in the Ca$^{2+}$-driven PS1 conformational change. (**A**) Spectral FRET analysis of the PS1 conformation in 7 W cells transfected with WT G-PS1-R and pre-treated with 1 μM KT5720 (KT, PKA inhibitor, 16 hr), 20 μM SP600125 (SP, JNK inhibitor, 4 hr), 100 nM Ro 31–8220 (Ro, PKC inhibitor, 16 hr), or 5 μM TDZD-8 (TD, GSK3$\beta$ inhibitor, 4 hr). Spectral FRET imaging was performed after the cells were treated with DMSO (vehicle control) or 5 μM A23187 for 15 min (n = 31–50 cells). Mean ± SEM, *p<0.05, **p<0.01, ***p<0.001, one-way factorial ANOVA. (**B**) Endogenous

*Figure 3 continued on next page*

*Figure 3 continued*

PS1 conformation was monitored by the antibody-based FLIM analysis. Primary neurons were pre-treated with PKA inhibitors: 30 µM H-89 or 1 µM KT5720 for 16 hr, or with PKA activators: 10 µM forskolin or 0.5 mM 8-Bromo-cAMP for 1 hr, followed by the treatment with vehicle control (-) or 50 mM KCl (+) for 5 min. The FRET efficiency in each group is normalized to the average FRET efficiency of vehicle control treated neurons (n = 36–125 processes from n = 17–28 cells). Mean ± SEM, ***p<0.001, one-way factorial ANOVA. (C) $Ca^{2+}$ sequestration by EGTA affects PS1 conformation in vehicle but not PKA activator-treated neurons. Primary neurons were pre-treated with vehicle, 10 µM forskolin or 0.5 mM 8-Bromo-cAMP for 1 hr, followed by the treatment with 2 mM EGTA or vehicle control for 15 min. The FRET efficiency in each group is normalized to the average FRET efficiency of vehicle control neurons (n = 19–34 cells, n = 40–103 processes). Mean ± SEM, **p<0.01, ***p<0.001, one-way factorial ANOVA. (D) FLIM analysis of the PS1 conformation in 7 W cells transfected with WT or phosphorylation-inhibited mutants G-PS1-R, and treated with 10 µM forskolin or vehicle control for 1 hr. The FRET efficiency is normalized to the average FRET efficiency of vehicle-treated cells expressing WT G-PS1-R (n = 20–39 cells). Mean ± SEM, *p<0.05, **p<0.01, one-way factorial ANOVA. (E) FLIM analysis of the PS1 conformation in 7 W cells transfected with WT or domain 1 + 2 phosphorylation-mimicking G-PS1-R isoform, and treated with 5 µM A23187, 10 µM forskolin or 0.5 mM 8-Bromo-cAMP. The FRET efficiency normalized to that in vehicle treated cells is shown (n = 34–47 cells). Mean ± SEM, *p<0.05, **p<0.01, one-way factorial ANOVA. (F) KCl-induced increase of the Aβ42/40 ratio is prevented by PKA inhibitor. Primary neurons were pre-treated with vehicle control or 1 µM KT5720 for 16 hr, followed by the treatment with vehicle control or 50 mM KCl for 30 min. Aβ40 and Aβ42 levels were measured by ELISA. The Aβ42/40 ratio in each group is normalized to that of the vehicle control neurons (n = 19–25). Mean ± SEM, *p<0.05, **p<0.01, one-way factorial ANOVA.

The following figure supplement is available for figure 3:

**Figure supplement 1.** PKA is involved in the $Ca^{2+}$-triggered PS1 pathogenic 'closed' conformation.

G-PS1-R was significantly increased upon A23187, forskolin or 8-Bromo-cAMP treatments (*Figure 3E*). This indicates that PKA is involved in the phosphorylation of S365-S366-S367 residues.

Since allosteric changes in PS1 conformation underlie changes in the Aβ42/40 ratio, with the 'closed' conformation linked to the increased Aβ42/40 ratio (*Berezovska et al., 2005*; *Isoo et al., 2007*; *Uemura et al., 2010*; *Wahlster et al., 2013*; *Kuzuya et al., 2016*), we next tested if the inhibition of PKA activity would prevent the KCl treatment-induced increase in the Aβ42/40 ratio. We found that treatment with KT5720 indeed abolished the KCl-induced elevation of the Aβ42/40 ratio (*Figure 3F*).

Collectively, these data demonstrate that phosphorylation of domain 1 or 2 is not sufficient for the PS1 conformational change, but is required for PKA-mediated phosphorylation of domain 3, leading to the pathogenic conformational change in PS1/γ-secretase.

## PS1 phosphorylation is responsible for the pathogenic conformational shift in vivo

We established that PS1 phosphorylation at domain 3 forces PS1 into the 'closed' state in vitro. To determine whether $Ca^{2+}$ influx may trigger PS1 domain 3 phosphorylation and the pathogenic 'closed' conformation in vivo, we optimized our spectral FRET assay (*Uemura et al., 2009*) to monitor PS1 conformation in living mice using two-photon microscopy. First, we found that GFP emission was the strongest in the 875–900 nm wavelength range, whereas RFP emission significantly decreased at >800 nm excitation, with barely detectable RFP signal at 900 nm in 7 W cells expressing G-PS1-R (*Figure 4—figure supplement 1A*). Therefore, 900 nm excitation was used in subsequent in vivo experiments. Using two different negative controls for FRET: G-PS1 (donor only) and PS1-R (acceptor only), we ensured that the emission intensity of RFP at 900 nm excitation is caused neither by GFP fluorescence bleed-through, nor by direct RFP excitation (*Figure 4—figure supplement 1B*).

Next, to express the G-PS1-R probe in the brains of living mice, we sub-cloned the construct into an AAV8 vector under the neuron specific human Synapsin 1 (hSyn1) promoter (*Figure 4A*). pAAV8-hSyn1-G-PS1 (donor only) and pAAV8-hSyn1-PS1-R (acceptor only) as negative controls, and pAAV8-hSyn1-R-G fusion protein as a positive control were also generated for the FRET assay. G-PS1-R expression in neurons of the somatosensory cortex was verified by two-photon imaging using 775 nm wavelength by which both GFP and RFP were excited simultaneously (*Figure 4B*). We selectively excited GFP at 900 nm, and found that the RFP/GFP (R/G) ratio was significantly higher in G-PS1-R expressing neurons than that in G-PS1 (donor only) (*Figure 4C*), validating the FRET detection and establishment of the PS1 conformation assay in living mouse brain.

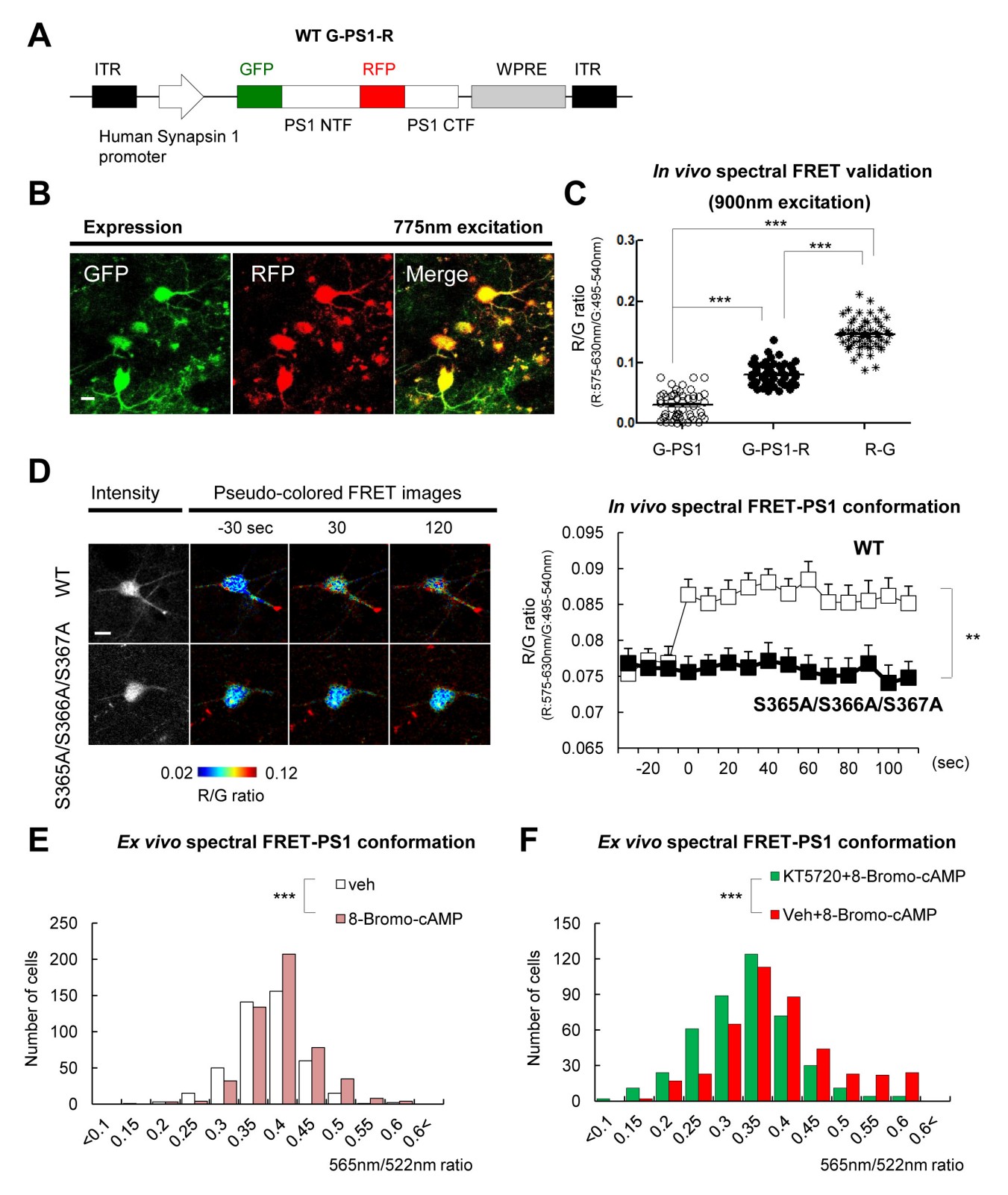

**Figure 4.** Ca$^{2+}$ triggers the PS1/γ-secretase pathogenic conformation via PS1 phosphorylation in vivo. (**A**) Schematic representation of the pAAV8-hSyn1-WT G-PS1-R construct. (**B**) Two-photon image of the WT G-PS1-R expression in the somatosensory cortex of WT mouse. Laser at 775 nm wavelength was used for the excitation. A scale bar indicates 10 μm. (**C**) Mice were injected with AAV8-hSyn1-G-PS1 (as a negative control of FRET), AAV8-hSyn1-WT G-PS1-R or AAV8-hSyn1-R-G (as a positive control of FRET). GFP was excited at 900 nm wavelength, and the R/G ratio was recorded

*Figure 4 continued on next page*

*Figure 4 continued*

(n = 50–60 cells, n = 3–6 mice). Mean ± SEM, ***p<0.001, one-way factorial ANOVA. (D) Spectral FRET analysis of the PS1 conformation in vivo. Mice were injected with AAV8-hSyn1-WT G-PS1-R (n = 3) or AAV8-hSyn1-S365A/S366A/S367A G-PS1-R (n = 4), and 300 mM KCl was applied topically. The R/G ratio in vivo was monitored after two-photon excitation at 900 nm (total n = 20–28 cells per condition) for the duration of 2 min. Representative images of the pseudo-colored neurons are shown (additional time traces/images are shown in *Figure 4—figure supplement 3*). Mean ± SEM, **p<0.01, two-way repeated-measures ANOVA. (E) Ex-vivo spectral FRET analysis of the endogenous PS1 conformation in mouse brain sections. Mice were injected with 100 mM 8-Bromo-cAMP (right hemisphere) or vehicle (left hemisphere) into the somatosensory cortex. The 565 nm/522 nm ratio was calculated in individual neurons as readout of the FRET efficiency that reflects the relative proximity of the PS1 NT (A488) to PS1 CT (Cy3). The histogram shows cell numbers plotted against the 565 nm/522 nm ratios. n = 3 mice, total of 444 (vehicle) and 505 (8 Bromo-cAMP) neurons. ***p<0.001, Student's t-test. (F) Ex-vivo spectral FRET analysis of the endogenous PS1 conformation in mouse brain sections. Mice were pre-injected with 100 μM KT5720 (right hemisphere) or vehicle (left hemisphere) into the somatosensory cortex. 75 min post-injection, 100 mM 8-Bromo-cAMP (both hemispheres) was delivered to the same area for 5 min. The histogram shows cell numbers plotted against the 565 nm/522 nm ratios. n = 3 mice, total 423 (vehicle) and 436 (KT5720) neurons analysed. ***p<0.001, Student's t-test.

The following figure supplements are available for figure 4:

**Figure supplement 1.** Establishment of the two-photon spectral FRET settings for monitoring PS1 conformation.

**Figure supplement 2.** YC3.6-based $Ca^{2+}$ imaging in vivo after KCl application.

**Figure supplement 3.** Spectral FRRT assay of PS1 conformation in vivo.

**Figure supplement 4.** Spectral FRET assay for monitoring endogenous PS1 conformation in mouse brain sections.

To ensure that the G-PS1-R probe was functional and able to reliably report PS1 conformational changes in vivo, we applied KCl topically, under the cranial window. First, we verified that topical KCl application significantly increased intracellular $Ca^{2+}$ levels in the somatosensory cortex neurons by employing Yellow Cameleon 3.6 (YC3.6)-based (*Nagai et al., 2004*) in vivo $Ca^{2+}$ imaging (*Figure 4—figure supplement 2*). The actual concentration of $Ca^{2+}$ in cell bodies, calculated as described previously (*Kuchibhotla et al., 2008*), was increased from ~58 nM at the basal condition up to ~2793 nM immediately after the applied KCl reached the imaging site. It remained at ~166 nM during the 2 min of imaging (mean = ~649 nM). Consistently, the R/G ratio was significantly enhanced by the KCl application (*Figure 4D* and *Figure 4—figure supplement 3*), indicating that elevated $Ca^{2+}$ induces the pathogenic PS1 conformation in vivo.

To confirm that $Ca^{2+}$ triggers the 'closed' conformational shift in PS1/γ-secretase via PS1 domain 3 phosphorylation, we expressed the phosphorylation-inhibited mutant S365A/S366A/S367A G-PS1-R in mouse brain prior to KCl stimulation. Notably, no change in the R/G ratio was detected in neurons expressing this mutant G-PS1-R after KCl application (*Figure 4D* and *Figure 4—figure supplement 3*). These findings indicate that increased intracellular $Ca^{2+}$ triggers the 'closed' conformation of PS1 via domain 3 phosphorylation in vivo.

To ensure that PKA-mediated phosphorylation is involved in the conformational change of the endogenous PS1, 8-Bromo-cAMP and vehicle control were injected into the somatosensory cortex of mice, in right and left hemispheres, respectively. The mice were sacrificed 5 min post-injection, and PKA activation was verified by immunostaining of the brain sections for p-CREB S133 phosphorylation (*Figure 4—figure supplement 4A*). Endogenous PS1 was immunolabeled with PS1 NT (A488) and PS1 CT (Cy3) antibodies, and the 565 nm/522 nm fluorescence emission ratio was used as a readout of the FRET efficiency (*Figure 4—figure supplement 4B*). The *ex-vivo* spectral FRET assay revealed that PKA activation by 8-Bromo-cAMP led to the 'closed' conformation of endogenous PS1 as indicated by the increased number of neurons with a higher 565 nm/522 nm ratio (*Figure 4E*). Next, to determine if KT5720 pre-treatment would prevent the Bromo-cAMP-induced pathogenic collapse of PS1, the PKA inhibitor KT5720 or vehicle control were injected into mouse somatosensory cortex 75 min prior to 8-Bromo-cAMP injection. The ex-vivo spectral FRET assay revealed that PKA inhibition could prevent the 8-Bromo-cAMP-triggered 'closed' conformation of PS1 in mouse brain (*Figure 4F*). Immunostaining for CREB S133 phosphorylation confirmed that KT5720 significantly suppressed 8-Bromo-cAMP-induced PKA activation (*Figure 4—figure supplement 4C*).

## PS1 phosphorylation is enhanced in the AD brain

Since PS1 adopts the pathogenic 'closed' conformation in sporadic AD brains (*Wahlster et al., 2013*), we investigated whether PS1 phosphorylation is up-regulated in the sAD brain. To test this, we used the commercially available S310 (domain 2) phosphorylation specific antibody to compare the amount of phosphorylated PS1 in AD brains and in age, gender and post mortem interval (PMI)-matched control brains (*Table 2*).

First, the specificity of the p-PS1 (S310) antibody to detect $Ca^{2+}$-induced PKA-mediated phosphorylation at the S310 residue was verified in 7 W cells. The cells were transfected with FLAG-tagged WT or S310A mutant PS1 constructs, treated with the calcium ionophore A23187, and phospho-PS1 (S310) immunoreactivity was detected by confocal microscopy (*Figure 5—figure supplement 1A*). We show that A23187 increased WT PS1 phosphorylation at S310 was abolished by pretreatment with the PKA inhibitor KT5720 or PS1 S310A expression (*Figure 5—figure supplement 1A*). Increased PS1 S310 phosphorylation by the PKA activator forskolin was also detected by Western blotting in PS70 cells (*Figure 5—figure supplement 1B*). In addition, we verified dose-dependent S310 phosphorylated PS1 (p-PS1) in human brain lysates by loading variable amounts of the total protein (*Figure 5—figure supplement 1B*), suggesting that the amount of PS1 phosphorylation is measurable in human brains. We found that even though a broad distribution of the p-PS1(S310) level is observed among the AD cases, the average PS1 phosphorylation is significantly increased in AD brains as compared to non-demented control brains (*Figure 5*). Correlation analysis showed that neither age nor PMI significantly affected the level of PS1 phosphorylation in both control and AD cases (*Figure 5—figure supplement 2A and B*).

## Discussion

Increasing evidence suggests interactions between $Ca^{2+}$ overload and A$\beta$ in the pathogenesis of AD. Although it has been suggested that aberrant $Ca^{2+}$ homeostasis affects A$\beta$ production, the effect of $Ca^{2+}$ elevations on PS1/$\gamma$-secretase function is unknown. We have recently reported that increases in $Ca^{2+}$ levels induce the pathogenic change in PS1 conformation and increase the A$\beta$42/40 ratio in cultured neurons (*Kuzuya et al., 2016*). Considering the rapid and dynamic nature of the conformational changes, it is unlikely that this change is achieved through mechanisms involving transcription or reassembly of PS1 into de novo $\gamma$-secretase complexes. However, the detailed molecular mechanism remains unclear, and whether similar $Ca^{2+}$-mediated PS1 conformational changes occur in vivo is unknown. Our current study addresses these questions and demonstrates that $Ca^{2+}$/PKA-mediated PS1 phosphorylation at domain 1 (T74), domain 2 (S310, S313) and domain 3 (S365, S366, S367) is responsible for the PS1/$\gamma$-secretase pathogenic conformational change in vitro and in vivo.

The high level of serines and threonines in the PS1 N-terminus and in the large hydrophilic loop region suggests that the enzymatic function of PS1 can be modulated by its 'phosphorylated' and 'dephosphorylated' states. It has been reported that PS1 phosphorylation at several residues enhances stability of the PS1 CT fragment that is necessary for $\gamma$-secretase activity (*Lau et al., 2002*; *Kuo et al., 2008*; *Ryu et al., 2010*). Recently, Matz and colleagues have identified eleven new phosphorylation sites in PS1 (*Matz et al., 2015*). They showed that phosphorylation-inhibited mutations of these sites affect neither activity of the PS1/$\gamma$-secretase nor A$\beta$ production. Our current study revealed that six amino acids: T74, S310, S313, S365, S366 and S367, all, except for S310, within the newly-discovered PS1 phosphorylation sites, are critical for the $Ca^{2+}$-triggered PS1 pathogenic, 'closed' conformation linked to increased A$\beta$42/40 ratio (*Berezovska et al., 2005*; *Isoo et al., 2007*; *Uemura et al., 2010*; *Wahlster et al., 2013*; *Kuzuya et al., 2016*). We found that the conformation of the phosphorylation-inhibited PS1 mutants at these residues is not different from that of the WT PS1. This is consistent with the findings of unaltered A$\beta$42/40 ratio in cells expressing PS1 phosphorylation-inhibited mutants (*Matz et al., 2015*). However, the conformation of the phosphorylation-inhibiting mutants and WT PS1 is very different in the $Ca^{2+}$ influx-stimulated condition. This finding suggests that phosphorylation of PS1 at these residues as a result of the $Ca^{2+}$ influx/PKA activation underlies its conformational change.

Our data show that introducing phosphorylation-inhibited mutations within all six residues prevents the $Ca^{2+}$ influx/PKA activation-triggered PS1 conformational change. This indicates that phosphorylation of domain 1 (T74), domain 2 (S310-S313) and domain 3 (S365-S366-S367) is important

**Table 2.** List of the human brain samples used in the study.

| Case | Age | PMI | Sex | Braak | Cerad |
|---|---|---|---|---|---|
| Control 1 | 60 | 15 | F | | |
| Control 2 | 73 | 20 | F | | |
| Control 3 | 88 | 20 | F | II | |
| Control 4 | 91 | 8 | F | I | A |
| Control 5 | 63 | 16 | M | | |
| Control 6 | 85 | 8 | M | I | |
| Control 7 | 87 | 48 | M | I | |
| Control 8 | 91 | 19 | F | II | A |
| Control 9 | 86 | 10 | M | | |
| Control 10 | 94 | 17 | M | I | |
| Control 11 | 54 | 6 | M | | |
| Control 12 | 58 | 18 | F | | |
| Control 13 | 88 | 16 | F | II | |
| Control 14 | 60 | 14 | M | | |
| Control 15 | 92 | unknown | M | II | |
| Control 16 | 68 | 27 | M | I | |
| Control 17 | 76 | 48 | F | I | possibly A |
| Control 18 | 92 | 12 | M | II | |
| Control 19 | 85 | unknown | M | II | |
| Control 20 | 92 | 23 | M | II | A |
| Control average (Mean ± SEM) | 79.15 ± 3.0 | 19.16 ± 2.6 | (F:M = 8:12) | | |
| AD 1 | 58 | 12 | F | VI | C |
| AD 2 | 73 | 18 | F | V | C |
| AD 3 | 84 | 24 | F | VI | C |
| AD 4 | 85 | 5 | F | VI | C |
| AD 5 | 60 | 24 | M | VI | C |
| AD 6 | 78 | 18 | F | VI | C |
| AD 7 | 85 | 24 | M | VI | C |
| AD 8 | 86 | 20 | M | VI | C |
| AD 9 | 87 | 4 | F | VI | C |
| AD 10 | 69 | 5 | M | VI | C |
| AD 11 | 94 | 12 | F | VI | C |
| AD 12 | 86 | 12 | M | VI | B |
| AD 13 | 89 | 18 | F | VI | B |
| AD 14 | 71 | 16 | F | VI | C |
| AD 15 | 70 | 17 | M | VI | C |
| AD 16 | 96 | 18 | M | VI | C |
| AD 17 | 91 | 6 | M | VI | C |
| AD 18 | 83 | 14 | F | VI | C |
| AD 19 | 87 | 12 | F | IV | B |
| AD 20 | 87 | 13 | M | VI | C |
| AD 21 | 73 | 14 | M | VI | C |
| AD 22 | 78 | 8 | M | VI | C |

*Table 2 continued on next page*

*Table 2 continued*

| Case | Age | PMI | Sex | Braak | Cerad |
|---|---|---|---|---|---|
| AD 23 | 79 | 8 | M | VI | C |
| AD average (Mean ± SEM) | 80.39 ± 2.1 | 14 ± 1.3 | (F:M = 11:12) | | |

for PS1 adopting the 'closed' conformation. On the other hand, the results using phosphorylation-mimicking mutants revealed that phosphorylation of domain 1 and domain 2, though necessary, is not sufficient for the structural change of PS1. Whereas domain 3, specifically S367, is critical for PS1 achieving the 'closed' conformation. These results beg the questions of how domain 3 is phosphorylated and whether phosphorylation of domain 1 and domain 2 is involved in the $Ca^{2+}$-triggered PS1 pathogenic conformational change. Although the domain 1 and domain two phosphorylation-

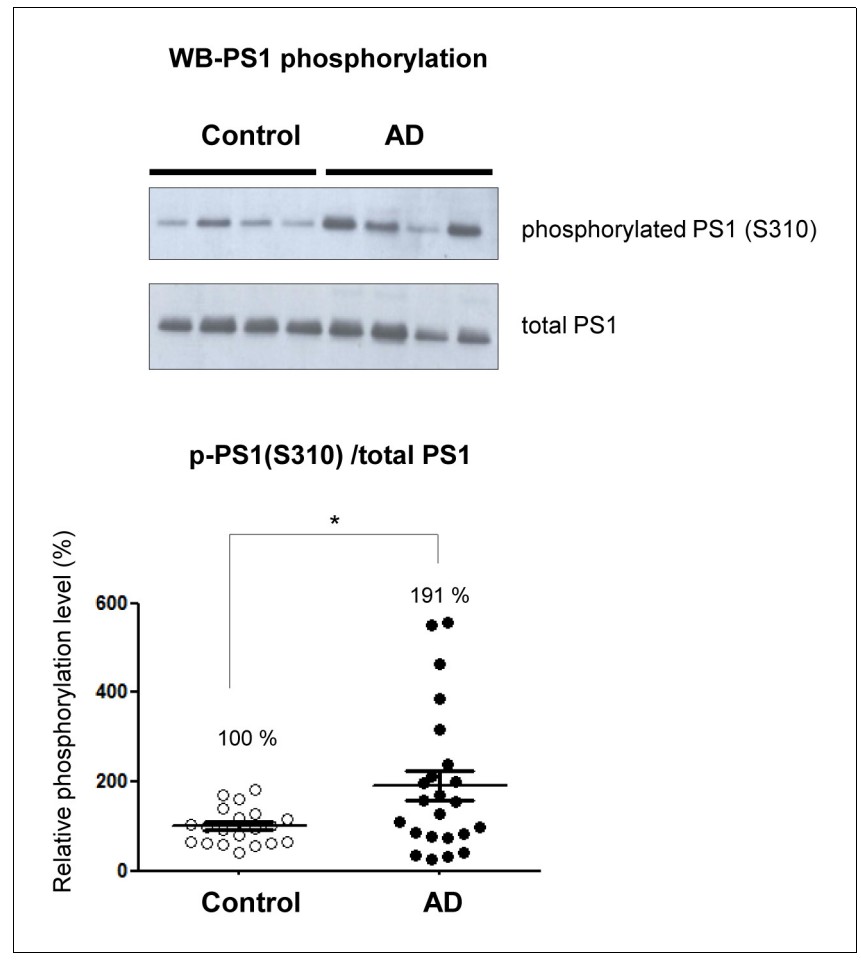

**Figure 5.** Phosphorylated PS1 level is increased in AD brains. Western blot analysis of PS1 phosphorylated at S310 in 20 control brains and 23 AD brains. The level of PS1 S310 phosphorylation (phospho-PS1/total PS1) in AD brains is normalized to that in the control brains. Mean ± SEM, *$p<0.05$, Student's t-test.

The following figure supplements are available for figure 5:

**Figure supplement 1.** Validation of the PKA-mediated PS1 phosphorylation at S310.

**Figure supplement 2.** Correlation between the relative phosphorylation level of PS1 and age, or PMI.

mimicking PS1 isoform (T74D/S310D/S313D G-PS1-R) shows the same conformation as WT G-PS1-R, PKA activators were able to induce the 'closed' conformation of this mutant. Therefore, we conclude that PKA is involved in phosphorylation of domain 3. We propose the following model: PKA first phosphorylates PS1 at domain 2, which leads to spatial rearrangements around domain 1, and allows its phosphorylation. This subsequently changes the local conformation around domain 3, enabling its phosphorylation by PKA. This final phosphorylation of S365-S366-S367, with S367 being most crucial, drives the pathogenic conformational alteration of the PS1/γ-secretase (*Figure 6—figure supplement 1A*). We propose that local spatial rearrangement around domain three is a key step for PKA-induced domain three phosphorylation, and phosphorylation of domain 1 and domain 2 is necessary for this to occur. This model is consistent with the finding that PKA is able to phosphorylate only S310 within domain 2 in recombinant PS1 hydrophilic loop (263–407) peptide in vitro (*Fluhrer et al., 2004*). We reason that lack of domain 1 in this recombinant PS1 fragment does not allow change in the local conformation around domain 3, explaining why S365-S366-S367 are not phosphorylated by PKA in this recombinant PS1. Moreover, a recent structural simulation study has demonstrated that 350–373 residues of PS1 containing domain 3, are subject to major movements (*Somavarapu and Kepp, 2016*). It is highly likely that naturally occurring domain 1 and domain 2 phosphorylation events may trigger these local spatial rearrangements around the 350–373 residues of PS1. We confirmed that domain 1 or domain 3 phosphorylation-inhibited mutant PS1 were still able to get phosphorylated by PKA at domain 2 (*Figure 6—figure supplement 1B*), suggesting that domain 2 is the first step of the phosphorylation. Consistently, except for domain 2, there are no known PKA substrate consensus sequences (*Ubersax and Ferrell, 2007*) around domain 1 or domain 3 of PS1 (*Figure 6—figure supplement 1C*).

PS1 conformational shifts are reflected by consistent changes in the FRET efficiency, detected by both the antibody-based FLIM assay of endogenous PS1 and by spectral FRET using the G-PS1-R reporter. Of note, all FRET-based assays only report relative changes in the proximity between the fluorophores, which does not necessarily translate into comparable changes in the distance between the protein epitopes (PS1 NT-CT and NT-loop, in this case). However, regardless of which assay we use, increased $Ca^{2+}$/PKA activation reliably and reproducibly increased the proximity between donor and acceptor fluorophores tagging PS1 NT-CT or NT-loop domains. This strongly suggests that we observe conformational changes in the entire PS1 molecule rather than just a change in the orientation of the donor and acceptor fluorophores.

We have also established and validated novel in vivo PS1 conformation assay. Topical KCl application significantly increased intracellular $Ca^{2+}$ (mean = ~649 nM, ranging from 166 nM to 2793 nM) in WT mice brains, a concentration of $Ca^{2+}$ consistent with that observed in AD mouse models displaying $Ca^{2+}$ overload in vivo (mean = ~499 nM, ranging from 162 nM to 1535 nM) (*Kuchibhotla et al., 2008*). We verified that $Ca^{2+}$ influx induces the pathogenic 'closed' PS1 conformation in vivo via PS1 S365/S366/S367 phosphorylation. The cAMP-triggered PKA activation in mouse brain and consecutive conformational change of the wild type PS1 that we report confirms PKA-mediated PS1 phosphorylation at endogenous levels. More importantly, we show up-regulation of the domain 2 (S310) phosphorylation in AD brains, demonstrating that $Ca^{2+}$-driven PS1 phosphorylation at certain residues may be important factors contributing to the pathogenesis of AD. Of note, a wide distribution of the phosphorylated PS1 levels observed in AD cases did not correlate with either age or PMI. Since we have selected a relatively homogenous group of AD brains in terms of pathology, Braak and CERAD scores could not be used to explain the wide distribution. Further investigation is necessary to determine if other factors, such as medications taken, disease duration, ApoE genotype, etc. could have affected the p-PS1 levels.

We propose a positive feed-forward mechanism for the pathogenesis of AD (*Figure 6*): once AD pathology, i.e. Aβ oligomers or deposited Aβ, triggers $Ca^{2+}$ overload, domains 1, 2 and 3 of PS1 can be hyper-phosphorylated by PKA. This induces the pathogenic 'closed' PS1 conformation, and leads to subsequent elevation of the Aβ42/40 ratio, followed by accelerated Aβ oligomerization or deposition. In line with these findings, we have previously reported that PS1 shows a 'closed' conformation around Aβ plaques in the sporadic AD brain (*Wahlster et al., 2013*), and that higher Aβ42/40 ratio leads to the pathogenic conformation of PS1/γ-secretase (*Zoltowska et al., 2016*). It is conceivable that selective inhibition of PS1 phosphorylation at the domain 1, 2 and/or 3 may present a novel opportunity for therapeutics for AD. Since phosphorylation-inhibited PS1 mutations on these sites do not affect assembly, maturation, or activity of the PS1/γ-secretase (*Matz et al., 2015*),

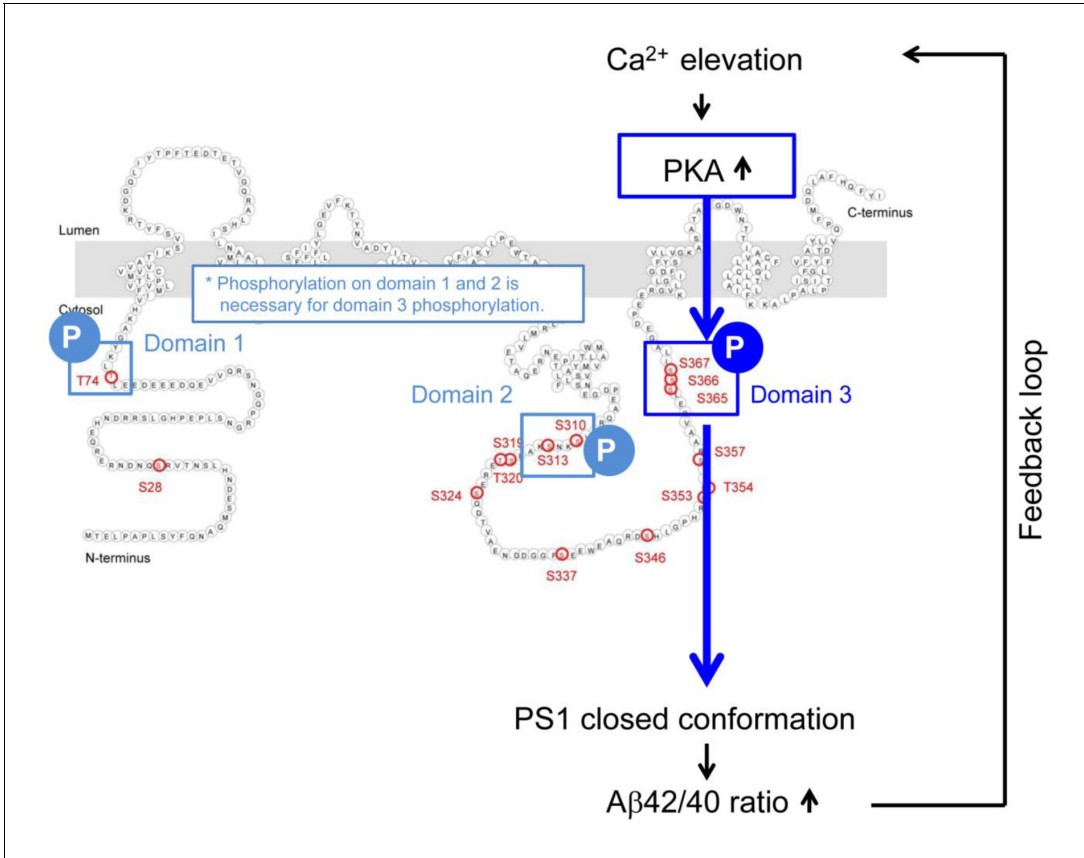

**Figure 6.** Mechanism of the Ca²⁺-triggered PS1 pathogenic conformational change. The schematic image of the molecular events involved in the Ca²⁺-triggered pathogenic 'closed' conformational change and increase of the Aβ42/40 ratio. The elevated Ca²⁺ levels induce PKA activation, followed by the phosphorylation of PS1 at domain 1, domain 2 and domain 3. Domain 3 phosphorylation, particularly at S367, induces the PS1 pathogenic conformation that leads to increase in the Aβ42/40 ratio.

The following figure supplement is available for figure 6:

**Figure supplement 1.** Model of the PKA-mediated PS1 phosphorylation.

inhibition of these PS1 phosphorylation residues would affect neither the cleavage of Notch nor other γ-secretase substrates. On the other hand, inhibition of PKA activity may not be a suitable approach to prevent PS1 phosphorylation, due to multiple PKA substrates and potential off target effects in the brain and in the periphery. In future studies, PS1-targeting agents that selectively block PS1 phosphorylation at either one of the three residues could be identified, and their efficacy could be examined in vivo using our novel PS1 conformation assays.

## Materials and methods

### Plasmid constructs

The N-terminally FLAG-tagged wild type (WT) PS1 and the N-terminally FLAG-tagged phosphorylation-inhibited PS1 mutants: S28A PS1, T74A PS1, S310A/S313A PS1, S313A PS1, S324A PS1, S337A PS1, S365A PS1, S366A PS1, S367A PS1, S365A/S367A PS1 and S366A/S367A PS1, were kind gifts from Dr. Patrick C Fraering (Brain Mind Institute and School of Life Sciences, Ecole Polytechnique Fédérale de Lausanne, Lausanne, Switzerland) (*Matz et al., 2015*). The N-terminally FLAG-tagged phosphorylation-inhibited PS1 mutants; S310A PS1, S319A/T320A PS1, S346A PS1, S353A PS1, T354A PS1, S357A PS1 and S353A/S357A PS1, were generated from the N-terminally FLAG-tagged WT PS1 by site-directed mutagenesis using the QuikChange Site-directed mutagenesis kit (Agilent

Technologies, Santa Clara, CA, USA). Primers; S310A PS1: 5′-GCTCAAAGGAGAGTAGCCAAAAA TTCCAAGTAT-3′ and 5′-ATACTTGGAATTTTTGGCTACTCTCCTTTGAGC-3′, S319A/T320A PS1: 5′-AAGTATAATGCAGAAGCCGCAGAAAGGGAGTCA-3′ and 5′-TGACTCCCTTTCTGCGGCTTCTGCA TTATACTT-3′, S346A PS1: 5′-GAAGCCCAGAGGGACGCTCATCTAGGGCCTCATCGC-3′ and 5′-GCGATGAGGCCCTAGATGAGCGTCCCTCTGGGCTTC-3′, S353A PS1: 5′-CTAGGGCCTCA TCGCGCTACACCTGAGTCACGA-3′ and 5′-TCGTGACTCAGGTGTAGCGCGATGAGGCCCTAG-3′, T354A PS1: 5′-CCTCATCGCTCTGCACCTGAGTCACGA-3′ and 5′-TCGTGACTCAGGTGCA-GAGCGATGAGG-3′, S357A PS1: 5′-CTACACCTGAGGCACGAGCTGCTGTCC-3′ and 5′-GGACAG-CAGCTCGTGCCTCAGGTGTAG-3′, S353A/S357A PS1:5′-CCTCATCGCGCTACACC TGAGGCACGAGCTGCT-3′ and 5′-AGCAGCTCGTGCCTCAGGTGTAGCGCGATGAGG-3′. The phosphorylation-inhibited PS1 mutants; T74A GFP-PS1-RFP (G-PS1-R), S310A/S313A G-PS1-R and S365A/S366A/S367A G-PS1-R, and the phosphorylation-mimicking PS1 mutants; T74D G-PS1-R, S310D/S313D G-PS1-R, S365D/S366D/S367D G-PS1-R, S365D G-PS1-R, S366D G-PS1-R and S367D G-PS1-R were generated from the PS1 conformation sensitive FRET probe, G-PS1-R, encoding WT PS1 with enhanced green fluorescent protein (GFP) fused to the NT and red fluorescent protein (RFP) inserted into the cytoplasmic loop (*Uemura et al., 2009*) by site-directed mutagenesis using the QuikChange Site-directed mutagenesis kit. Primers; T74A G-PS1-R: 5′-GAAGATGAGGAGC TGGCATTGAAATATGGCGCC-3′ and 5′-GGCGCCATATTTCAATGCCAGCTCCTCATCTTC-3′, S310A/S313A G-PS1-R: 5′-AGGAGAGTAGCCAAAAATGCCAAGTATAATGCA-3′ and 5′-TGCATTA TACTTGGCATTTTTGGCTACTCTCCT-3′, S365A/S366A/S367A G-PS1-R: 5′-GTCCAGGAAC TTGCCGCCGCTATCCTCGCTGGT-3′ and 5′-ACCAGCGAGGATAGCGGCGGCAAGTTCCTGGAC-3′. Primers; T74D G-PS1-R: 5′-GAAGATGAGGAGCTGGACTTGAAATATGGCGCC-3′ and 5′-GGCGCCATATTTCAAGTCCAGCTCCTCATCTTC-3′, S310D/S313D G-PS1-R: 5′-GCTCAAAGGA-GAGTAGACAAAAATGACAAGTAT-3′ and 5′-ATACTTGTCATTTTTGTCTACTCTCCTTTGAGC-3′, S365D/S366D/S367D G-PS1-R: 5′-GTCCAGGAACTTGACGACGATATCCTCGCTGGT-3′ and 5′-ACCAGCGAGGATATCGTCGTCAAGTTCCTGGAC-3′. S365D G-PS1-R: 5′- GCTGTCCAGGAAC TTGACAGCAGTATCCTCGCT-3′ and 5′- AGCGAGGATACTGCTGTCAAGTTCCTGGACAGC-3′. S366D G-PS1-R: 5′- GTCCAGGAACTTTCCGACAGTATCCTCGCTGGT-3′ and 5′- ACCAGCGAGGA TACTGTCGGAAAGTTCCTGGAC-3′. S367D G-PS1-R: 5′- CAGGAACTTTCCAGCGATATCCTCGC TGGTGAA-3′ and 5′- TTCACCAGCGAGGATATCGCTGGAAAGTTCCTG-3′. T74D/S310D/S313D G-PS1-R was generated from S310D/S313D G-PS1-R, T74D/S365D/S366D/S367D G-PS1-R was from S365D/S366D/S367D G-PS1-R, S310D/S313D/S365D/S366D/S367D G-PS1-R was from S365D/ S366D/S367D G-PS1-R and T74D/S310D/S313D/S365D/S366D/S367D G-PS1-R was from S310D/ S313D/S365D/S366D/S367D G-PS1-R using primer sets shown above. The constructs with GFP fused to the NT of PS1 (G-PS1, donor only), RFP inserted into the large hydrophilic loop domain of PS1 (PS1-R acceptor only), and RFP-GFP fusion (R-G fusion) plasmid, in which RFP is fused to the NT of GFP with a short linker, were used as controls for the FRET assays (*Uemura et al., 2009*). R-G324D-(Kin-8), a dominant negative form of the PKA regulatory subunit α, was obtained from Addgene (plasmid # 28179) (*Olson et al., 1993*).

## Antibodies and chemical reagents

The mouse monoclonal anti-PS1 NT (RRID:AB_301867), rabbit monoclonal anti-PS1 loop (RRID:AB_1310605), rabbit monoclonal anti-phosphorylated PS1 at S310 (RRID:AB_1267317), and mouse monoclonal anti-phosphoserine (RRID:AB_305611) antibodies were purchased from Abcam (Cambridge, MA, USA). The rabbit polyclonal anti-PS1 CT (RRID:AB_261178) and mouse monoclonal anti-FLAG (RRID:AB_259529) antibodies were from Sigma-Aldrich (St. Louis, MO, USA). The mouse monoclonal anti-PS1 loop (RRID:AB_95175), rabbit polyclonal anti-phosphoserine (RRID:AB_390205) and rabbit polyclonal anti-phosphorylated CREB at S133 (RRID:AB_310153) antibodies were from Millipore (Temecula, CA, USA). Rabbit monoclonal CREB antibody (RRID:AB_309979) was from Upstate Biotechnology (Lake Placid, NY, USA). Mouse (RRID:AB_737182) and rabbit (RRID:AB_737197) normal IgG were from Santa Cruz Biotechnology, Inc. (Dallas, TX, USA). Alexa Fluor 488 (A488), and Cy3-labeled corresponding secondary antibodies were from Life Technologies (Grand Island, NY, USA) and horseradish peroxidase (HRP)-conjugated secondary antibodies were from Pierce (Rockford, IL, USA). The Ca$^{2+}$ ionophore A23187, PKA inhibitors H-89 and KT5720, PKA activators forskolin and 8-Bromoadenosine 3′:5′-cyclic monophosphate (8-Bromo-cAMP), JNK inhibitor

SP600125, PKC inhibitor Ro 31–8220, and GSK3$\beta$ inhibitor TDZD-8 were from Sigma-Aldrich (St. Louis, MO, USA).

## Cell culture and transfection

CHO cell lines stably overexpressing WT APP (7 W cells) (*Koo and Squazzo, 1994*) and stably overexpressing WT APP and WT PS1 (PS70 cells) (*Xia et al., 1997*) were a kind gift from Dr. Dennis Selkoe (Brigham and Women's Hospital, Harvard Medical School, USA). PS1/PS2 double knockout mouse embryonic fibroblasts (PS1/2 dKO MEF) were a kind gift from Dr. Bart De Strooper (Katholieke Universiteit Leuven, Leuven, Belgium). These cells were authenticated using STR profiling, monitored for mycoplasma contamination every 2 months, and maintained as described previously (*Uemura et al., 2010*). Lipofectamine 3000 (Life technologies, Carlsbad, CA, USA) was used for transient transfection according to the manufacturer's instructions. Primary neuronal cultures were obtained from cerebral cortex and hippocampus of mouse embryos at gestation day 14–16 (Charles River Laboratories, Wilmington, MA). The neurons were dissociated using Papain Dissociation System (Worthington Biochemical Corporation, Lakewood, NJ, USA) and were maintained for 13–15 days in vitro (DIV) in Neurobasal medium containing 2% B27 supplement, 1% GlutaMax, and 1% Pen/Strep mix (Life Technologies).

## AAV preparation and virus injection

All experimental procedures using mice were in compliance with the NIH guidelines for the use of animals in experiments and were approved by the Massachusetts General Hospital Animal Care and Use Committee. In this study, we developed five constructs that were packaged into viruses: WT G-PS1-R (pAAV8-human Synapsin1 (hSyn1)-WT G-PS1-R), S365A/S366A/S367A G-PS1-R (pAAV8-hSyn1-S365A/S366A/S367A G-PS1-R), G-PS1 (pAAV8-hSyn1-G-PS1), PS1-R (pAAV8-hSyn1-PS1-R) and RFP-GFP fusion (pAAV8-hSyn1-R-G). These plasmids were constructed by cloning corresponding PS1 constructs into an AAV8 vector containing the hSyn1 promoter and the woodchuck hepatitis virus posttranscriptional regulatory element (WPRE) sequences, using Nhel/HindIII or HindIII/EcoRI restriction sites. All plasmids were packaged in AAV serotype 8 at the University of Pennsylvania Vector Core. Viral titers for all constructs were between $4.9 \times 10^{12}$ GC/ml to $1.5 \times 10^{13}$ GC/ml. The construction of pAAV2-CBA-Yellow Cameleon 3.6 (YC3.6) was described previously (*Kuchibhotla et al., 2008*).

Experimental procedures for viral injection have been described previously (*Kuchibhotla et al., 2008*). Briefly, five to seven months old male C57BL/6 mice (Charles River Laboratories, Wilmington, MA, USA) were used for intra-cortical injection of the viruses. Mice were anesthetized with 1–3% isoflurane and placed in a stereotaxic apparatus. The body temperature of mice was maintained with a heating system throughout the procedures. The surgical region was sterilized with isopropyl alcohol and betadyne, and burr holes were drilled in the skull. Viruses were injected into the somatosensory cortex (2 mm lateral, 1.5 mm posterior to Bregma, 0.7 mm deep) using a Hamilton syringe from Sigma-Aldrich (St. Louis, MO, USA) at a speed of 130 nl/min. After injection, the skin was sutured and treated with Fougera Triple Antibiotic Ointment (E. FOUGERA and CO., Melville, NY, USA).

## Cranial window surgery

One month after viral injection, mice were anesthetized with 1–3% isoflurane and placed in a stereotaxic apparatus. A circular area of skull was carefully drilled, removed, and then dura was carefully removed. A circular cranial window soaked in PBS was implanted to the brain surface, and fixed to the remaining skull with dental cement mixed with Krazy glue. For topical KCl application, a burr hole was created in the fixed dental cement to inject KCl with a Hamilton syringe into the space between the cranial window and brain surface.

## Drug injection

Five to seven months old male C57BL/6 mice were anesthetized as described above, placed into the stereotax, and injected with 1 µl of 100 mM 8-Bromo-cAMP (right hemisphere) or vehicle (left hemisphere) into the somatosensory cortex. 5 min post-injection, mice were sacrificed using $CO_2$ asphyxiation, perfused intracardially with PBS followed by 4% PFA. For the rescue experiments, 1 µl of 100

µM KT5720 (right hemisphere) or vehicle (left hemisphere) was injected 75 min prior to 8-Bromo-cAMP treatment.

## Immunohisto/cytochemistry for antibody-based FRET assays

The brains were dissected, post-fixed by immersion in 4% paraformaldehyde (PFA) with 15% glycerol (Sigma-Aldrich, St, Louis, MO) in PBS, and cryoprotected using 30% glycerol in PBS. Prior to immunostaining, the brains were cut on a freezing sliding microtome (Leica SM 2000R, Bannockburn, IL) into 35 µm-thick coronal sections. Brain tissue sections were permeabilized using 0.4% TX-100. For 4% PFA fixed 7 W cells and primary neurons 0.1% TX-100 was used. Non-specific binding of the antibodies was blocked by incubation of the cultured cells or free-floating brain sections with 1.5% normal donkey serum (Jackson ImmunoResearch Labs, West Grove, PA). The primary antibodies were applied overnight at 4°C, followed by 1 hr incubation at room temperature with corresponding Alexa Fluor 488- or Cy3-conjugated secondary antibodies. The samples were mounted in VectaShield mounting medium (Vector Laboratories, Inc., Burlingame, CA).

## Fluorescence lifetime imaging microscopy (FLIM)

The FLIM analysis of PS1 conformation was performed as described previously (*Berezovska et al., 2005*). Briefly, a mode-locked Chameleon Ti:Sapphire laser (Coherent Inc., Santa Clara, CA) set at 850 nm or 770 nm was used to excite GFP or Alexa Fluor 488 donor fluorophores, respectively. A Zeiss LSM510 microscope and x63 oil immersion objective was used for the imaging. The donor fluorophore lifetimes were recorded using a high-speed photomultiplier tube (MCP R3809; Hamamatsu, Bridgewater, NJ) and a time-correlated single-photon counting acquisition board (SPC-830; Becker and Hickl, Berlin, Germany). The baseline lifetime ($t1$) of the donor fluorophore was measured in the absence of the acceptor fluorophore (negative control, FRET absent). In the presence of the acceptor fluorophore (RFP or Cy3), excitation of the donor fluorophore results in reduced donor emission energy if the donor and acceptor are less than 5–10 nm apart (FRET present). This yields characteristic shortening of the donor fluorophore lifetime ($t2$). The acquired FLIM data were analyzed using SPC Image software (Becker and Hickl, Berlin, Germany) by fitting the data to one (negative control) or two (acceptor present) exponential decay curves. In the latter case, $t1$ of the non-FRETing population was 'fixed' and thus excluded from the analysis, and only shorter, $t2$, values were analyzed. The FRET efficiency ($\%E_{FRET}$) was calculated using the following equation: $\%E_{FRET} = 100*(t1-t2)/t1$. Higher $\%E_{FRET}$ reflects closer proximity between fluorophores labeling the PS1 domains.

## Spectral FRET

The spectral FRET assay with single photon excitation for the experiments using cultured cells and immunostained mouse brain sections was conducted as described previously (*Uemura et al., 2009*). Briefly, an Argon laser at 488 nm was used to excite GFP or Alexa 488, and emitted fluorescence was detected by seven channels of the Zeiss Metadetector within the 502–651 nm or 511–682 nm wavelength range (21.4 nm spectral bandwidth for each channel) on a Zeiss LSM510 microscope. Average pixel fluorescence intensity for the whole cell after subtraction of the background fluorescence was measured using Image J. The ratio of fluorescence intensity in the 598 nm channel (for RFP) to that in the 513 nm channel (for GFP) or 565 nm (Cy3) to 522 nm (Alexa 488) was used as a readout of the FRET efficiency, which reflects the relative proximity between the donor and acceptor.

The spectral FRET assay for monitoring PS1 conformation in living mouse brain using two-photon excitation is newly established. First, to determine the excitation wavelength that preferentially excites GFP, the G-PS1-R probe was excited at different wavelengths from 750 nm to 975 nm with a mode-locked titanium/sapphire laser (MaiTai; Spectra-Physics, Fremont, CA). The 900 nm wavelength was chosen to selectively excite GFP, and emitted fluorescence was detected by two emission channels: 495–540 nm range for channel 1 (for GFP) and 575–630 nm for channel 2 (for RFP), on an Olympus Fluoview 1000 MPE microscope (x20 objective, water immersion, NA = 1.05) (Olympus Corporation, Tokyo, Japan). Time-lapse images were obtained every 10 s for a duration of 2 min. The average pixel fluorescence intensity after subtraction of the background fluorescence for the whole cell was measured using ImageJ in each channel. The R/G ratio was used as readout of the FRET efficiency. Pseudo-colored images were generated in MATLAB.

## Ca$^{2+}$ imaging

Intracellular Ca$^{2+}$ levels in 7 W cells were determined using the ratiometric Ca$^{2+}$-sensitive dye Indo-1 (*Grynkiewicz et al., 1985*). Briefly, Indo-1/AM (Thermo Fisher Scientific, Inc., Cambridge, MA) was dissolved with 20% pluronic F-127 (Thermo Fisher Scientific, Inc.) in DMSO and added to the culture dishes at a final concentration of 1 µM Indo-1/AM and 0.02% pluronic F-27 for 45 min. Images were obtained using a Zeiss LSM510 microscope (x25 water immersion objective, Ca$^{2+}$/Mg$^{2+}$ containing PBS, 37°C, 5% CO$_2$). A Chameleon Ti:Sapphire laser was used for excitation at 750 nm, and the emitted fluorescence was detected in two channels: 390–465 nm and 500–550 nm.

Intraneuronal Ca$^{2+}$ levels in the somatosensory cortex of living mice was measured using the FRET-based ratiometric probe, Yellow Cameleon 3.6 (YC3.6) (*Nagai et al., 2004*), as described previously (*Kuchibhotla et al., 2008*). Briefly, a mode-locked titanium/sapphire laser (MaiTai; Spectra-Physics, Fremont, CA) at 860 nm was used for excitation of CFP, and the emitted fluorescence was detected in two channels: 460–500 nm (for CFP) and 520–560 nm (for YFP) (x20 objective). Time-lapse images were obtained every 10 s for 2 min. Pseudo-colored images were generated in MAT-LAB. The ratiometric analysis of the YFP/CFP ratio and quantitative determination of the [Ca$^{2+}$] concentrations was conducted as described previously (*Kuchibhotla et al., 2008*).

## Measurement of extracellular Aβ

The levels of secreted Aβ40 and Aβ42 were measured using ELISA kits according to the manufacturer's instructions (Wako, Osaka, Japan).

## Immunoprecipitation and western blotting

Primary neurons were lysed in radio-immunoprecipitation assay (RIPA) buffer with protease and phosphatase inhibitor cocktail (Fisher Scientific, Pittsburg, PA, USA). The cells were homogenized and incubated for 30 min on ice. Each sample was then centrifuged, and the supernatants were collected. Protein concentrations were determined using a Pierce BCA Protein Assay Kit (Thermo Fisher Scientific, Inc.). Aliquots with equal amount of total protein were treated with protein G-Sepharose (GE Healthcare Bio-Sciences, Uppsala, Sweden) for 1 hr. After removing protein G-Sepharose by centrifugation, primary antibodies were added to the supernatants. Each sample was rotated for 1 hr and then treated with protein G-Sepharose for 1 hr. The immunoprecipitates were washed with RIPA buffer and resuspended in 2 x LDS sample buffer (Life Technologies). After boiling and centrifugation, the supernatants were subjected to SDS-PAGE on NuPAGE Bis-Tris gels (Life Technologies).

Frontal cortex of frozen human brains from neuropathologically verified AD cases were obtained from the brain bank of the Alzheimer's Disease Research Centre (ADRC) at Massachusetts General Hospital, and were lysed by RIPA buffer with protease and phosphatase inhibitor cocktail (Fisher Scientific) and calyculin A (Cell Signaling Technology, Danvers, MA, USA). Control cases were non-demented individuals who did not meet pathological diagnostic criteria of AD or any other neurodegenerative diseases (*Table 2*). All the subjects or their next of kin gave informed consent for the brain donation.

## Statistical analysis

Statistical analysis was performed using StatView for Windows, Version 5.0.1 or GraphPad Prism 5. All values are given as means ± SEM. Comparisons were performed using One sample t-test, Student's t-test, or one-way factorial ANOVA followed by Bonferroni's post-hoc analysis. For the in vivo time course experiments monitoring PS1 conformation and Ca$^{2+}$ concentration, two-way repeated-measures ANOVA and Bonferroni post-hoc analysis were used. For the correlation analysis in the human brain-based experiments, Spearman's nonparametric correlation analysis was used. An appropriate sample size was computed when the study was being designed and each experiment was performed using at least three independent trials. $p < 0.05$ was considered to indicate a significant difference.

## Acknowledgements

We are grateful to Dr Patrick C Fraering (Brain Mind Institute and School of Life Sciences, Ecole Polytechnique Fédérale de Lausanne, Switzerland) for sharing the PS1 constructs, to Ms Hannah R

Weisman (Massachusetts General Hospital) for help with IHC/ex-vivo spectral FRET analysis, to Ms Zhanyun Fan (Massachusetts General Hospital) for help with plasmids/AAV construction, to Drs Michal Arbel-Ornath and Steven S Hou (Massachusetts General Hospital) for help with YC3.6-based $Ca^{2+}$ imaging/analysis, to Dr Steven Rodriguez (Massachusetts General Hospital) for help with human brain sample preparation and to Dr Shuko Takeda (Massachusetts General Hospital) for helpful discussion. This work was supported by the National Institute of Health grants AG 044486 (OB) and AG 15379 (OB), the Kanae Foundation for the Promotion of Medical Science grant (MM), the Japan Society for the Promotion of Science fellowship (MM), the Kyoto University Foundation fellowship (MM), the German National Academic Foundation fellowship (JH) and the EMBO Long-Term Fellowship (ATLF-815-2014), which is co-funded by the Marie Curie Actions of the European Commission (LTFCOFUND2013, GA-2013-609409) (EK).

## Additional information

### Funding

| Funder | Grant reference number | Author |
| --- | --- | --- |
| Kanae Foundation for Promotion of Medical Science | The 43rd Kanae Foreign Study Award, 2014-55 | Masato Maesako |
| Japan Society for the Promotion of Science | Postdoctoral Fellowship for Research Abroad | Masato Maesako |
| Kyoto University Education and Research Foundation | Postdoctoral Fellowship for Research Abroad | Masato Maesako |
| Studienstiftung des Deutschen Volkes | Medical student Scholarship | Jana Horlacher |
| European Molecular Biology Organization | ATLF-815-20, Postdoctoral Long Term Fellowship co-funded with Marie Curie Actions of the European Commission | Eleanna Kara |
| European Commission | Marie Curie Actions, LTFCOFUND2013, GA-2013-609409 | Eleanna Kara |
| NIH Blueprint for Neuroscience Research | AG15379 | Oksana Berezovska |
| NIH Blueprint for Neuroscience Research | AG44486 | Oksana Berezovska |

The funders had no role in study design, data collection and interpretation, or the decision to submit the work for publication.

### Author contributions

MM, Acquisition of data; Analysis and interpretation of data; Drafting or revising the article; JH, EK, SS, LJK, Acquisition of data, Analysis and interpretation of data; KMZ, KVK, Acquisition of data, Contributed unpublished essential data or reagents; XL, BTH, Conception and design, Analysis and interpretation of data; BJB, Conception and design, Analysis and interpretation of data, Drafting or revising the article; OB, Conception and design; Acquisition of data; Analysis and interpretation of data; Drafting or revising the article

### Author ORCIDs

Masato Maesako, http://orcid.org/0000-0002-1970-2462
Katarzyna M Zoltowska, http://orcid.org/0000-0001-5853-3465
Oksana Berezovska, http://orcid.org/0000-0003-4898-5788

### Ethics

Human subjects: Frontal cortex of frozen human brains were obtained from the brain bank of the Alzheimer's Disease Research Centre (ADRC) at Massachusetts General Hospital. All the subjects or their next of kin gave informed consent for the brain donation.

Animal experimentation: All experimental procedures using mice were in compliance with the NIH guidelines for the use of animals in experiments and were approved by the Massachusetts General Hospital Animal Care and Use Committee (2016N000243). All surgery was performed under isoflurane anesthesia, and every effort was made to minimize suffering.

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
