## [Decision Letter]

Thank you for submitting your article "Pathogenic PS1 phosphorylation at Ser367" for consideration by *eLife*. Your article has been favorably evaluated by a Senior Editor and two reviewers, one of whom is a member of our Board of Reviewing Editors. The reviewers have opted to remain anonymous.

The reviewers have discussed the reviews with one another and the Reviewing Editor has drafted this decision to help you prepare a revised submission.

The manuscript by Maesako et al. reports on a very interesting set of studies of the link between calcium increases, PS1 phosphorylation, enhancing the population of its closed conformation, along with pathogenic increases in the Abeta42 to Abeta40 ratio.

As the authors can see from consulting both reviews, the reviewers want to see the raw FRET data central to your main conclusions and better understand your longer wavelength antibody-based FRET assay in fixed tissue. In the traditional FRET assay the changes are small and both reviewers want to be convinced that these changes are significant. In addition, reviewer 1 wants to see your pharmacologic line of evidence expanded into at least two cell lines and ideally in mice to provide a line of evidence to support your hypothesis in addition to your PS1 Ser/Thr to Ala mutations and Ser/Thr to Asp mutational data.

Please also consider the reviews pasted below in full in terms of revising your manuscript.

*Reviewer #1:*

This manuscript is generally well written and the experimental execution is earnest.

This is a complicated paper experimentally in the sense that it is assumed that the PS1 Ser/Thr to Ala mutations and Ser/Thr to Asp mutations have no influence other than preventing and simulating phosphorylation, respectively –the desired interpretation. Of course, these mutations can change conformational ensemble for other reasons as well. Collectively, considering the data as a whole in this paper, it is supportive of the hypothesis reflected by the title.

That said, the changes observed for individual dataplots are small and in many cases only significant in some cases because of smaller than typical error bars on for example the control measurements (e.g. Figure 1 – central to the paper seemingly). Figure 3 is also unconvincing. The data is certainly trending in the right direction, and significant with the selected comparisons.

Unless I am missing something, why don't the authors use their new FRET assay (Figure 4) in cells, wherein the differences are easier to see and apparently are larger, as opposed to the data in Figure 3. Maybe this is typical of traditional FRET measurements, but their new FRET assay seems very promising in terms of quantifying the closed vs. open conformation. I am used to seeing histograms of closed and open FRET data, not interpreted data, but I will leave this up to the FRET experts to sort out.

I want to know more about KT5720, the pharmacologic PKA inhibitor-its selectivity, potency, etc. This is critical for me to get comfortable with publishing this paper. I want to see a second line of evidence to support the authors hypothesis (i.e. in addition to the PS1 Ser/Thr to Ala mutations and Ser/Thr to Asp mutations). If this inhibitor is selective, it seems that this is the most unassailable data in the paper to show via the suite of experiments in several cell lines and ideally in mouse brain that inhibitor pre-treatment prevents pathogenic collapse of PS1 and prevents altering the Abeta ratio associated with pathology, in the presence of triggered calcium increases. As presented this section is intriguing but not compelling.

There are modern mass spectrometry methods that also should prove useful in affinity purifying PS1 to ascertain its phosphorylation state quantitatively, not by reliance solely on the PS1 Ser/Thr to Ala mutations and Ser/Thr to Asp mutations. If the pharmacologic data is not possible to generate, then this type of data becomes central as a second line of evidence.

*Reviewer #2:*

The manuscript by Maesako et al. reports on a very interesting set of studies of the link between PS1 phosphorylation and its closed conformation, with links to AD. The authors use a series of experiments, including with biochemical and FRET experiments to systematically dissect the relationship of different phosphorylation states of the protein and its conformational properties in cells and neurons. The model that emerges is a complex one where Ca^2+^ triggered PKA activity results in sequential phosphorylation of domains 2, 1 and 3, with S367 phosphorylation being critical for change to a closed conformation. An exciting additional set of experiments directly probe FRET in vivo, supporting the developed model. Finally, an important link to disease was uncovered by the observation that the phosphorylation at S310 (domain 2) increased on average in AD brains. Overall, the work represents a strong combination of interesting tools with many important control experiments to develop a mechanistic model for how a PKA phosphorylation cascade results in PS1 conformational changes including in vivo. The obtained novel mechanistic understanding combined with the observed links of phosphorylation levels in AD suggests potential venues for therapeutic strategies. Therefore, I believe the paper would be of interest to the broad readership of *eLife*. Some technical issues are noted below that should be addressed by the authors.

1) The antibody based FRET experiments need more description and discussion, at least in the Methods section.

How large are the antibodies and how are they labeled? What range of distance changes (at least approximately) are being detected? Are factors other than distance affecting the rather small changes in the reported changes in FRET efficiency (Figure 1)? Factors such as orientational effects or changes in chromophore photophysics (less likely here) can also contribute to observed FRET efficiencies.

The changes in average reported FRET efficiencies are small. However, from the images in Figure 1, some portions of the neurons may have larger changes than others. Could the authors comment on this aspect and whether a more detailed analysis is possible?

2) Some of the above points are more generally relevant for the FRET studies in the paper, and authors should discuss these issues in the paper.

3) It would be useful to have a few additional images/time traces (corresponding to Figure 4) in the supplementary information for the reader's benefit.

4) The authors observe the interesting result that phosphorylation at domain 2 (S310) is increased on average in AD brains. However, in addition, Figure 5 lower panel indicates that there is a wider distribution of phosphorylation ranging from lower to much higher values than for controls. Authors please comment on this issue and how it affects their conclusions.

---

## [Author Response]

*[…]*

*Reviewer #1:*

*This manuscript is generally well written and the experimental execution is earnest.*

*This is a complicated paper experimentally in the sense that it is assumed that the PS1 Ser/Thr to Ala mutations and Ser/Thr to Asp mutations have no influence other than preventing and simulating phosphorylation, respectively –the desired interpretation. Of course, these mutations can change conformational ensemble for other reasons as well. Collectively, considering the data as a whole in this paper, it is supportive of the hypothesis reflected by the title.*

*That said, the changes observed for individual dataplots are small and in many cases only significant in some cases because of smaller than typical error bars on for example the control measurements (e.g. Figure 1 – central to the paper seemingly). Figure 3 is also unconvincing. The data is certainly trending in the right direction, and significant with the selected comparisons.*

We agree with the reviewer’s concern that mutations in PS1 phosphorylation sites could have changed the conformational ensemble of the PS1/γ-secretase. Thus, we have thoroughly analyzed conformation of the phosphorylation-inhibited Ser/Thr to Ala PS1 mutants and found that it is not different from that of the wild type PS1 in the basal, vehicle-treated, condition (Table 1 and Figure 2—figure supplement 1). This indicates that Ser/Thr to Ala mutations per se do not alter PS1/γ-secretase conformation. The effect of the phospho-mimicking Ser/Thr to Asp mutations, on the other hand, is more “multifaceted”, with domain 1 and domain 2 Asp mutants having no effect on PS1 conformation in the basal condition, but Ser to Asp domain 3 mutant shows the pathogenic conformation similar to that of drug-stimulated (KCl/A23187/ forskolin/cAMP) wild type PS1 expressing cells (Figure 2 and Figure 3). Together, this indicates that phosphorylation of PS1 at domain 3, and Ser 367 in particular, rather than mutation itself, is crucial for the pathogenic conformational change.

With regard to the small changes for individual data-plots (Figure 1 and Figure 3) and “smaller than typical error bars on, for example, the control measurements (e.g. Figure 1)”: First of all, we thank the reviewer for noticing this mistake and want to apologize for the oversight on our part: the graphing software generated error bars for the vehicle treated control in Figure 1 equivalent to “0”, and thus shouldn’t be shown at all. The statistical difference for Figure 1 was calculated based on the assumption that the mean in KCl treated cells is different from 100% (one-sample t-test). Specifically, we have performed six independent experiments of IP/Western blotting to compare the amount of phosphorylated PS1 between vehicle and KCl^-^treated primary neurons. To be able to pull these experiments together for the statistical analysis (and to avoid great variability in the band’s optical density due to different film exposures among different experiments), 100% was assigned to the vehicle treated sample, and the percent change in phospho-PS1 for KCl treated neurons was analyzed in each experiment. Phosphorylation is a very dynamic process and its detection can be influenced by a number of factors. However, even though the average increase in phospho-PS1 after KCl treatment was relatively small, the increase was consistent and reproducible. For the Aβ ELISA measurements (Figure 3),we have conducted five independent experiments, each containing n= 3 to 8 biological replicates (not just triplicates-octuplets), and thus error bars were generated for both vehicle and drug-treated samples. Again, even though the KCl^-^triggered Aβ42/40 ratio increase was relatively small it was consistent and significantly higher than that in control cells. The results from all trial are combined to show this.

*Unless I am missing something, why don't the authors use their new FRET assay (Figure 4) in cells, wherein the differences are easier to see and apparently are larger, as opposed to the data in Figure 3. Maybe this is typical of traditional FRET measurements, but their new FRET assay seems very promising in terms of quantifying the closed vs. open conformation. I am used to seeing histograms of closed and open FRET data, not interpreted data, but I will leave this up to the FRET experts to sort out.*

We appreciate the reviewer’s acknowledgment of the great potential of the new G-PS1-R FRET probe-based assay. The assay is indeed uniquely suited for the longitudinal real-time measurements of the dynamic PS1 conformational change in live cells in vitro and in vivo. However, to measure conformational change of the endogenous PS1 in neurons, the complementary antibody-based FLIM approach is needed.

To further substantiate the interchangeable application of these two assays, we have now performed a spectral FRET assay in live neurons in vitro infected with AAV/G-PS1-R, and found that KCl treatment leads to an increase in the R/G ratio (PS1 adopts “closed” conformation), whereas the PKA inhibitor KT5720 blocks this effect (new Figure 3—figure supplement 1). This result matches closely the results from the antibody-based FLIM assay monitoring changes in endogenous PS1 (Figure 3).

*I want to know more about KT5720, the pharmacologic PKA inhibitor-its selectivity, potency, etc. This is critical for me to get comfortable with publishing this paper. I want to see a second line of evidence to support the authors hypothesis (i.e. in addition to the PS1 Ser/Thr to Ala mutations and Ser/Thr to Asp mutations). If this inhibitor is selective, it seems that this is the most unassailable data in the paper to show via the suite of experiments in several cell lines and ideally in mouse brain that inhibitor pre-treatment prevents pathogenic collapse of PS1 and prevents altering the Abeta ratio associated with pathology, in the presence of triggered calcium increases. As presented this section is intriguing but not compelling.*

*There are modern mass spectrometry methods that also should prove useful in affinity purifying PS1 to ascertain its phosphorylation state quantitatively, not by reliance solely on the PS1 Ser/Thr to Ala mutations and Ser/Thr to Asp mutations. If the pharmacologic data is not possible to generate, then this type of data becomes central as a second line of evidence.*

We agree with the reviewer that having PKA-selective inhibitor(s) would provide truly unassailable data. Unfortunately, it is practically impossible to have a pharmacological reagent that is 100% selective. KT5720 has been reported/used in almost 1000 studies as a selective PKA inhibitor. Still, according to the review by Andrew Murry, (2008, Sci Signal), KT5720 may inhibit some other kinases as well, although to a lesser degree. Thus, we have tested several reported inhibitors of other kinases (i.e., *JNK*, PKC, and GSK3β) and found that unlike KT5720 inhibition of PKA, inhibition of these kinases was not able to prevent the Ca^2+^ influx-induced PS1 conformational change(Figure 3). To further corroborate the conclusion of the involvement of PKA, we have also used a different “preferred” PKA inhibitor, H-89, and two different PKA activators, forskolin and 8-Bromo-cAMP, and obtained similar results (Figure 3 and Figure 3—figure supplement 1). Together, these data strongly suggest that the observed effect of phosphorylation on PS1 conformational change is PKA-mediated.

However, to further address the reviewer’s concern about selective PKA role, and to verify the results from pharmacological manipulation, we have performed additional experiments using a different, “genetic”, approach: expression of the dominant negative form of the PKA regulatory subunit α. We found that both the Ca^2+^ influx-induced PS1 conformational change and PS1 S310 phosphorylation are significantly inhibited by the expression of the dominant negative PKA mutant vs. empty vector control in 7W CHO cells (new Figure 3—figure supplement 1). Based on all the data, we cannot fully exclude that other kinase(s) may also contribute to the PS1 conformation-inducing phosphorylation, however we are confident that activation of PKA is involved.

We appreciate the reviewer’s suggestion to expand our experiments to “several cell lines and ideally mouse brain”. We have performed the FLIM analysis in 7W line (CHO cells), and show that both PKA inhibitors, KT5720 and H-89, inhibit the Ca^2+^ influx-induced PS1 conformational change (new Figure 3—figure supplement 1). This is consistent with the results in primary neurons (Figure 3).

Furthermore, to verify the physiological relevance of the PKA-inhibition in mouse brain and to determine if KT5720 pre-treatment would prevent the cAMP-induced “pathogenic collapse of PS1”, we injected 1µl of 100 µM KT5720 inhibitor or vehicle control into the somatosensory cortex. 75 minutes post-injection, 1µl of 100 mM 8-Bromo-cAMP, a direct PKA activator, was delivered to the same brain region for 5 minutes. Mice were quickly euthanized, the brain was removed, fixed, sectioned, and ex-vivospectral FRET analysis of endogenous PS1 conformation was performed. We found that KT5720 prevents the 8-Bromo-cAMP-induced PS1 “closed” conformational change in mouse brain in vivo (new Figure 4). The inhibitory effect of KT5720 on PKA activity has been verified by immunostaining for PKA specific substrate, phospho-CREB (S133), in adjacent mouse brain sections (new Figure 4—figure supplement 4).

*Reviewer #2: […] 1) The antibody based FRET experiments need more description and discussion, at least in the Methods section.*

*How large are the antibodies and how are they labeled? What range of distance changes (at least approximately) are being detected? Are factors other than distance affecting the rather small changes in the reported changes in FRET efficiency (Figure 1)? Factors such as orientational effects or changes in chromophore photophysics (less likely here) can also contribute to observed FRET efficiencies.*

*The changes in average reported FRET efficiencies are small. However, from the images in Figure 1, some portions of the neurons may have larger changes than others. Could the authors comment on this aspect and whether a more detailed analysis is possible?*

*2) Some of the above points are more generally relevant for the FRET studies in the paper, and authors should discuss these issues in the paper.*

Per reviewer’s request we provided more details for the antibody-based FRET assays in the Material and methods of the revised manuscript. We appreciate and agree with the reviewer’s concern about large complexes of the primary + fluorescently labeled secondary antibodies. However, the great advantage of using the antibody-based FRET assay is the ability to monitor changes in the proximity between endogenous PS1/γ-secretase domains. We are aware that all FRET-based assays only report relative changes in the proximity between fluorophores tagging these domains, which does not directly translate into comparable changes in the distance between the protein epitopes (PS1 NT and CT, in this case). Nevertheless, both spectral FRET and FLIM assays we used have proven to be highly reliable and reproducible in reporting consistent changes in the FRET efficiency (reflecting PS1 conformational changes) after experimental treatments.

However, to further substantiate the findings we have also used our previously validated GFP-PS1-RFP FRET reporter probe (Uemura et al., 2009, Plos One) to monitor PS1 conformational change in real time in live cells in vitro and in vivo. In the G-PS1-R probe, donor and acceptor fluorophores are fused to the PS1 NT and cytosolic loop, respectively, reporting relative change in the GFP-NT to RFP-loop proximity.

Regardless of which assay we used, increased Ca^2+^/PKA activation consistently increases the proximity between donor and acceptor fluorophores tagging PS1 NT and CT in the antibody-based FRET (Figure 1, Table 1, Figure 3,new Figure 3—figure supplement 1)or PS1 NT and Loop domain in the G-PS1-R probe-based FRET assay (Figure 2—figure supplement 1, Figure 3,new Figure 3—figure supplement 1). This indicates that the antibody-based FRET assay reliably reflects PS1 conformational change. Moreover, since increased Ca^2+^ shortens both the PS1 NT-CT and PS1 NT-loop proximity, we believe we observe conformational change in the entire PS1 molecule rather than just a change in the orientation of the donor and acceptor fluorophores. Thus, whether it is KCl (Figure 1, Figure 3, Figure 2—figure supplement 2) or A23187 (Table 1, Figure 3, Figure 2—figure supplement 1, Figure 3—figure supplement 1) or PKA activation by forskolin and cAMP (Figure 3), and whether it is spectral FRET or the FLIM assay, we consistently see increases in the FRET efficiency reflecting increased proximity between phosphorylated PS1 domains –> PS1 adopting a “closed” conformation.

The average reported change in FRET efficiency is ~10-20% by spectral FRET vs. 30-50% by FLIM (more sensitive detection as it is fluorophore concentration independent). Even though the change in the proximity between fluorophores may be a relatively small, it could signal substantial conformational change on the molecular level that would affect PS1/γ-secretase function.

In regard to the range of the distance changes: Since in our GFP-PS1-RFP conformational reporter probe donor and acceptor are in the 1:1 ratio, and the Förster radius of the RFP-GFP pair is known (5.7 Å) (Shcherbo et al., 2009, BMC Biotechno), we could translate the FRET signal into the change in the distance between GFP and RFP by using the following equation: d= 5.7 x [1/(1-t2/t1) -1]^(1/6) (with t1 and t2 being donor fluorophore lifetimes in the absence and presence of the FRET acceptor, respectively). The GFP-NT/RFP-loop distance is approximately 70 Å in the vehicle-treated condition. Ca^2+^ stimulation (A23187, KCl) leads to the average per neuron change in the ~3~4.5 Å range.

We fully agree with the reviewer that some areas in neurons show higher FRET efficiency than others, which could be due to local differences in the Ca^2+^ concentration or in PKA distribution. A combination of the PS1 conformational assay and Ca^2+^ imaging or PKA staining in the same cell could provide more detailed mechanistic insight. However, since both rely on the fluorescent labeling/detection, our experience shows that the presence of the third fluorophore may interfere with the reliability of the FRET detection in cells.

*3) It would be useful to have a few additional images/time traces (corresponding to Figure 4) in the supplementary information for the reader's benefit.*

Per the reviewer’s request we have included additional images/time traces in the new Figure 4—figure supplement 3.

*4) The authors observe the interesting result that phosphorylation at domain 2 (S310) is increased on average in AD brains. However, in addition, Figure 5 lower panel indicates that there is a wider distribution of phosphorylation ranging from lower to much higher values than for controls. Authors please comment on this issue and how it affects their conclusions.*

Indeed, a wide distribution in the level of PS1 S310 phosphorylation was revealed in AD brains. It is possible thatbroad phospho-PS1distribution could be result of the phosphorylation dyshomeostasis in the distressed AD brains (i.e. over activation/inhibition of kinases/phosphatases). There are so many factors that are difficult to control or monitor (or simply have all information unavailable) when analyze human brain samples, such as individual progress of the disease, past medical history, medications, or even lifestyles, etc. All of these could contribute to broader variability in AD brain.

All AD and control cases we used in this study were matched as best as possible by the subject’s age and post-mortem interval (PMI). We have also selected a relatively homogenous group of AD patients in terms of pathology (almost all cases are B&B VI and CERAD C). Since the original submission of the manuscript we have analyzed more AD and Control cases, bringing the total number to n=20 Controls and n=23 AD. The wide distribution of phospho-PS1 among AD, but not Control, cases still remains. Having higher “n” allowed us to do correlation analysis with the available data. We found that neither age nor PMI affect the level of PS1 phosphorylation (updated Figure 5 and new Figure 5—figure supplement 2).

The reviewer raised the interesting question that warrants further investigation of the different influencing factors (e.g., ApoE genotype, disease duration/age of onset, etc.) that may shed light on variations in the PS1 S310 phosphorylation level between different AD subjects. Unfortunately, such data were only available for a very few of the AD cases used in this study, and thus cannot be included in this manuscript. Since the main focus of the current study is on the molecular mechanism of the pathogenic PS1 conformational change, a different follow up epidemiological study that will select a different set and significantly higher number of AD and Control cases with more complete clinical data would be needed.